# Liquid crystalline inverted lipid phases encapsulating siRNA enhance lipid nanoparticle mediated transfection

Roy Pattipeiluhu [1,2,4], Ye Zeng[1], Marco M.R.M. Hendrix[3], Ilja K. Voets [3], Alexander Kros [1] ✉ & Thomas H. Sharp [2,5] ✉

Efficient cytosolic delivery of RNA molecules remains a formidable barrier for RNA therapeutic strategies. Lipid nanoparticles (LNPs) serve as state-of-the-art carriers that can deliver RNA molecules intracellularly, as exemplified by the recent implementation of several vaccines against SARS-CoV-2. Using a bottom-up rational design approach, we assemble LNPs that contain programmable lipid phases encapsulating small interfering RNA (siRNA). A combination of cryogenic transmission electron microscopy, cryogenic electron tomography and small-angle X-ray scattering reveals that we can form inverse hexagonal structures, which are present in a liquid crystalline nature within the LNP core. Comparison with lamellar LNPs reveals that the presence of inverse hexagonal phases enhances the intracellular silencing efficiency over lamellar structures. We then demonstrate that lamellar LNPs exhibit an in situ transition from a lamellar to inverse hexagonal phase upon interaction with anionic membranes, whereas LNPs containing pre-programmed liquid crystalline hexagonal phases bypass this transition for a more efficient one-step delivery mechanism, explaining the increased silencing effect. This rational design of LNPs with defined lipid structures aids in the understanding of the nano-bio interface and adds substantial value for LNP design, optimization and use.

RNA therapy relies on the delivery of exogenous (therapeutic) RNA molecules, such as messenger (mRNA) or small interfering RNA (siRNA), to control disease-relevant gene expression[1–3]. For efficient functional cytosolic delivery to, and release within, target cells, these highly charged, immunogenic and membrane-impermeable RNA molecules require the use of delivery systems[4–6]. To this end, ionizable lipid nanoparticles (LNPs) serve as state-of-the-art vehicles that can package, protect and release RNA molecules inside cells[7,8]. LNPs have realized the translation of RNA therapeutics to the clinic, highlighted by the approval of Onpattro®[9], enabling RNA interference (RNAi) therapy for the treatment of polyneuropathies resulting from

transthyretin-mediated amyloidosis[10]. In addition, this platform has been successfully expanded for the delivery of other RNA molecules, yielding safe and effective mRNA-based vaccines for SARS-CoV-2[11,12].

LNPs are multicomponent systems typically composed of an ionizable lipid (IL), phospholipid, cholesterol, PEG-lipid and an nucleic acid payload, formulated through rapid microfluidic mixing[13,14]. At a pH below its pKa, the ionizable lipids enable electrostatic complexation of anionic nucleic acid molecules, followed by self-assembly and buffer exchange to physiological pH to form nanostructures with a core-shell structure in the range of ~30-150 nm[15]. The LNP core is considered "hydrophobic", being rich in ionizable lipids, cholesterol and its

[1]Supramolecular and Biomaterials Chemistry, Leiden Institute of Chemistry, Leiden University, Einsteinweg 55, 2333 CC Leiden, The Netherlands. [2]Department of Cell and Chemical Biology, Leiden University Medical Center, Einthovenweg 20, 2333 ZC Leiden, The Netherlands. [3]Self-Organizing Soft Matter, Department of Chemical Engineering and Chemistry & Institute of Complex Molecular Systems, Eindhoven University of Technology, P.O. Box 513, 5600 MB Eindhoven, The Netherlands. [4]Present address: BioNTech SE, An der Goldgrube 12, 55131 Mainz, Germany. [5]Present address: School of Biochemistry, University of Bristol, Bristol BS8 1TD, United Kingdom. ✉e-mail: a.kros@chem.leidenuniv.nl; t.sharp@bristol.ac.uk

nucleic acid payload, in contrast, the LNP surface (i.e. lipid-water interface) is rich in helper phospholipids and lipid-PEG conjugates[16]. Intracellular unpacking and delivery to the cytosol of the encapsulated RNA molecules relies on endosomal acidification and in situ protonation of the ILs, leading to an electrostatic interaction with the endosomal membrane[17]. Disruption of the LNP structure and endosomal membrane is crucial for sufficient cytosolic RNA delivery[18,19]. However, in this process the majority (≥98%) of RNA molecules delivered with LNP systems remain trapped inside endosomal and lysosomal compartments, leading to degradation or efflux out of the cell[20,21]. To this end, empirical studies exploring the chemical space of LNP components, for example, the diversification of IL structures[22–27], or the variation of helper and PEG-lipids[28–32], has been pursued in order to improve LNP-mediated transfection efficiency. More recently, similar studies have been coupled with biophysical characterization of lipid structures in LNPs, aiming to understand the importance of LNP lipid organization and structure on their biological activity[33–37]. However, characterization and identification of defined lipid structures in LNPs, and the mechanistic understanding of how these structures affect LNP-endosome interaction and RNA delivery into the cytoplasm, remains elusive[38,39]. Nevertheless, a fundamental understanding of these mechanisms can aid the development of more potent LNP nanomedicines.

Here, we present bottom-up rational design of LNPs with defined lipid superstructures encapsulating siRNA, in order to accurately study their structure-activity relationships. Using cryogenic transmission electron microscopy (cryoTEM), cryogenic electron tomography (cryoET) and small angle X-ray scattering (SAXS), we identify and characterize defined lamellar, liquid crystalline inverted hexagonal or mixed lipid-RNA structures in the core of LNPs, and are able to successfully differentiate between empty lipid structures and those containing siRNA. By combining cryoTEM and SAXS, we gain insights into the thermal stability of these structures and demonstrate that thermally-stable liquid crystalline inverse hexagonal lipid phases encapsulating siRNA enhance the LNP-mediated transfection efficiency over lamellar variants. Our data supports the notion that lamellar LNP formulations induce an in situ transition from a lamellar to inverse hexagonal phase upon interaction with anionic membranes, whereas LNPs with pre-programmed inverse hexagonal phases can bypass this transition for a more efficient one-step delivery mechanism. We believe that our rational approach and biophysical characterization will yield new avenues for LNP design and development and can improve the efficiency of RNA therapeutics.

## Results and Discussion

### Design of defined lipid phases in ionizable lipid nanoparticles

The formation of lipid superstructures is dictated by the composition and ratio of different lipids with distinctive biophysical properties, such as transition temperature, charge, hydrophobicity and intrinsic curvature[40]. This is utilized in the field of nanomedicine in order to generate non-lamellar and crystalline lipid nanoparticles[41,42]. For example, the formation of inverted lipid structures is dependent on an abundance of lipids that possess a intrinsic negative ($R_0 < 0$) curvature, such as the lipid 1,2-dioleoyl-sn-glycero-3-phosphoethanolamine (DOPE) (Fig. 1a)[43]. Here, we aimed to rationally design LNP systems containing lamellar or inverted structures encapsulating siRNA in order to assess their biophysical properties and structure-activity relationship. We envisioned that generation of lamellar or inverted phases in the core of LNPs could be achieved by manipulation of the DOPE lipid content (mol%), substituting for cholesterol (Fig. 1b,c). We chose three different formulations, in which the molar ratios of the model ionizable lipid (IL) 1,2-dioleoyl-3-dimethylammonium-propane (DODAP) and PEG lipid 1,2-dimyristoyl-sn-glycero-3-phosphoethanolamine-N-methoxypolyethylene glycol-2000 (DMPE-PEG2000) were kept constant, while varying in DOPE content at 10, 30 and 49 mol%

(Fig. 1c, Supplementary Fig. 1). These formulations are referred to as 10PE-LNP, 30PE-LNP and 49PE-LNP respectively. The small interfering RNA (siRNA) molecule Patisiran®[10], used in the clinically approved LNP formulation Onpattro®[9], was chosen a model oligonucleotide cargo (Supplementary Table 1). Duplexed siRNA molecules for therapeutic purposes are short (~18–22 base pairs)[44], display high structural similarity and are less prone to structural deformation or the formation of secondary structures compared to mRNA[45], which aids in the formation of predictable and reproducible lipid-RNA structures. LNP formulations were assembled with different ionizable lipid (nitrogen, N) to siRNA ratios (phosphate, P) ratios, being: no RNA, NP = 6 and NP = 1 (noted as 10PE-LNP-noRNA, 10PE-LNP-NP6 or 10PE-LNP-NP1, etc.). A NP ratio of 6 is commonly applied for oligonucleotide payloads such as siRNA or mRNA in order to achieve a sufficient encapsulated dose with high encapsulation efficiency[46], whereas a NP ratio of 1 is known to have surpassed the maximum encapsulation capacity of RNA molecules in LNPs[47]. We performed a comparative analysis of these LNPs in order to determine the independent effects of RNA content and lipid composition on the formation of lipid-RNA superstructures. Conventional microfluidic mixing procedures were used to fabricate the LNPs (Supplementary Fig. 2). After dialysis, the hydrodynamic radius and surface charge of the LNPs were determined by dynamic light scattering (DLS) and zeta potential measurements, respectively (Supplementary Table 2). Formulations with an NP ratio of 6 or 1 showed the formation of 100-150 nm particles with a polydispersity index (PDI) < 0.120, in which particle size slightly increased concomitantly with DOPE content. Assembly of 30PE-LNP-noRNA and 49PE-LNP-noRNA showed formation of larger and more polydisperse particles. In all cases, the surface charge varied between −1 to −5 mV, indicating the formation of zwitterionic neutral LNPs in PBS buffer, reflecting the expected formal charge of the lipid combination at physiological pH.

In order to gain insights into the formation of lipid structures in these formulations, we combined SAXS as bulk method with cryoTEM imaging to assess the difference in nanoscale structures of the assembled LNPs. All 10PE-LNP SAXS profiles displayed a Bragg reflection at a scattering vector $\mathbf{q} \sim 0.1\,Å^{-1}$, which was most pronounced at NP1. This is a hallmark of the formation of a defined structure for all NP ratios and its amplification by an increase in siRNA content in the assembly (Fig. 1d). The associated cryoTEM revealed the formation of lamellar structures in all three formulations, in which 10PE-LNP-NP1 formed concentric circles extending to the LNP core (Black arrows, Fig. 1g, Supplementary Fig. 3a). For 10PE-LNP-NP6 and 10PE-LNP-noRNA, structure formation was limited to several lamellar rings on the periphery, surrounding an amorphous core (Supplementary Fig. 4). When dialysis was performed with a molecular weight cut-off (MWCO) below the size of the siRNA (MWCO = 10 kDa) to exchange buffer but retain free RNA for effective encapsulation efficiency (EE%) determination, an EE% of roughly 50% indicated successful saturation of encapsulation of 10PE-LNPs at a NP ratio of 1 (Supplementary Fig. 5a). Our data showing the amplification of lamellar structures and saturation of encapsulation are in line with previously reported LNPs with comparable lipid compositions and NP ratios[15,47].

When the DOPE content was increased to 30 mol% (30PE-LNP variants), the Bragg reflection shifted towards smaller $\mathbf{q}$ values, indicating transition away from lamellar structures (Fig. 1e). Two maxima around $q \sim 0.1\,Å^{-1}$ are distinguishable for 30PE-LNP-noRNA and 30PE-LNP-NP6, whereas 30PE-LNP-NP1 showed a broad shoulder instead, suggesting that multiple structures may coexist. For 30PE-LNP-NP1, cryoTEM revealed the formation of various repeating and distinguishable structures. Similar to 10PE-LNP-NP1, lamellar structures could be identified (Black arrows, Fig. 1h, Supplementary Fig. 3b) but, in addition, also non-lamellar structures were observed (White arrows, Fig. 1h). In the case of 30PE-LNP-NP6 and 30PE-LNP-noRNA, similar lamellar and non-lamellar structures were observed (Supplementary Fig. 6). However, following the trend of the 10PE-LNP variants, these

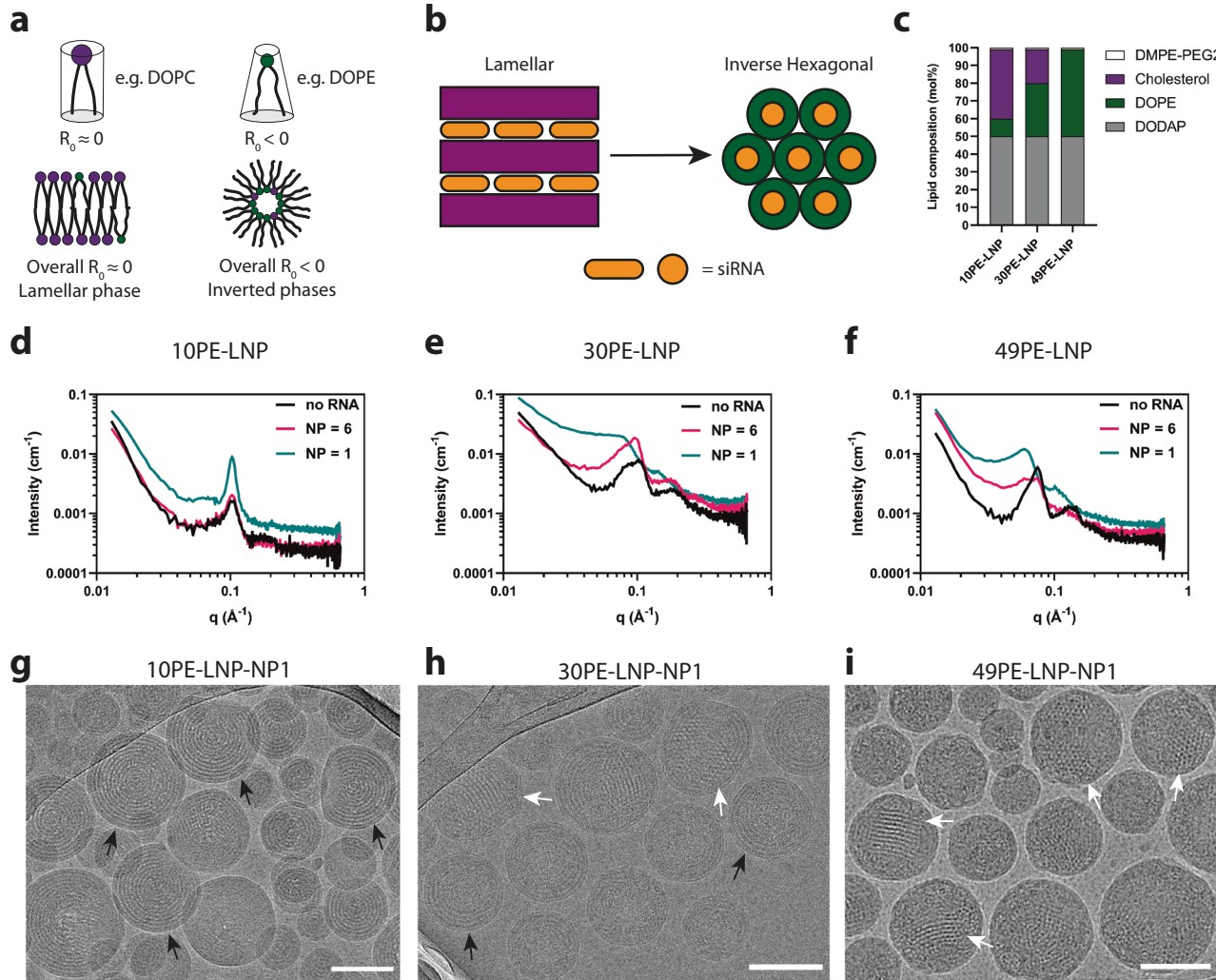

**Fig. 1 | Design of LNPs containing defined lipid phases. a** Curvature in lipid mixtures is driven by the composition of lipids with various curvature profiles Compositions with larger amounts of lipids with a $R_0 < 0$ lead to the formation of inverted phases. **b** Lamellar structures can transition towards inverted phases by the increase of DOPE lipid content. **c** LNP compositions designed to form lamellar and inverted lipid structures encapsulating siRNA used in this study. **d–f** SAXS profiles of the LNP compositions shown in **c**, at different N/P ratios. **g–i** Representative cryoTEM images of the LNPs compositions in **c**, at NP ratio = 1. Scale bars are 100 nm. Black arrows indicate the presence of lamellar structures. White arrows indicate the presence of nonlamellar structures. Micrographs are representative selections from a triplicate of experiments. Abbreviations used: NP Nitrogen to phosphate ratio. Source data are provided as a Source Data file.

structures were less defined and the lamellar rings showed a lower signal amplification in the presence of siRNA compared to those observed in 30PE-LNP-NP1.

When the DOPE content was increased further to 49 mol% (49PE-LNP variants), the $q \sim 0.1 \, \text{Å}^{-1}$ reflection again shifted to lower **q** values (Fig. 1f). In this case, both 49PE-LNP-NP1 and 49PE-LNP-noRNA showed discernable Bragg reflections. These occurred at lower **q** values for 49PE-LNP-NP1 than for 49PE-LNP-noRNA, indicating a difference in structure and/or domain size. The LNP-49PE-NP6 profile displays the principal reflections of both 49PE-LNP-NP1 and 49PE-LNP-noRNA, albeit less pronounced, indicating the formation of a mixture of the corresponding structures. In all of the LNP-49PE variants, cryoTEM revealed the sole formation of nonlamellar-like structures (Fig. 1i, Supplementary Fig. 3c and 7). Similar to 10PE-LNP-NP1, the 30PE-LNP-NP1 and 49PE-LNP-NP1 formulations displayed saturation of encapsulation, indicated by an EE% of roughly 50% (Supplementary Fig. 5). For all formulations, the large excess of siRNA could be removed by dialysis with a MWCO larger than free siRNA of 1 MDa, in order to avoid interference of unencapsulated siRNA in further experiments.

## Structural analysis of lamellar ($L_\alpha$) and liquid crystalline inverted hexagonal ($H_{II}$) phases in LNPs

Since 10PE-LNP, 30PE-LNP and 49PE-LNP assembled with an NP ratio of 1 formed particles of similar sizes, along with clearly defined lipid structures, they were subjected to comparative and in-depth structural analysis. For each of the formulations, a large number of single particles (n > 75) showing clear structural properties were selected from cryoTEM images and categorized based on structural similarity. The regions of interest were subjected to a fast Fourier transform (FFT), yielding the reciprocal lattice from which the repeating distance of a structure could be determined. In the case of 10PE-LNP-NP1, only lamellar concentric rings were observed throughout the sample. These structures displayed clear diffraction, as well as higher-order maxima, of the reciprocal space repeating unit, specifically at 6.38 nm, 3.19 nm and 2.08 nm (Fig. 2a). The initial scattering vector at 6.38 nm represents the lattice spacing of the primitive cell. Analyzing all of the selected single particles allowed us to construct a violin plot, revealing the variation in lamellar lattice spacing, which range from 4.20 nm to 6.90 nm (Fig. 2b). This range is not well reflected in the SAXS profile of 10PE-LNP-NP1, which displays a narrow peak at $q \sim 0.1 \, \text{Å}^{-1}$ (Fig. 1d).

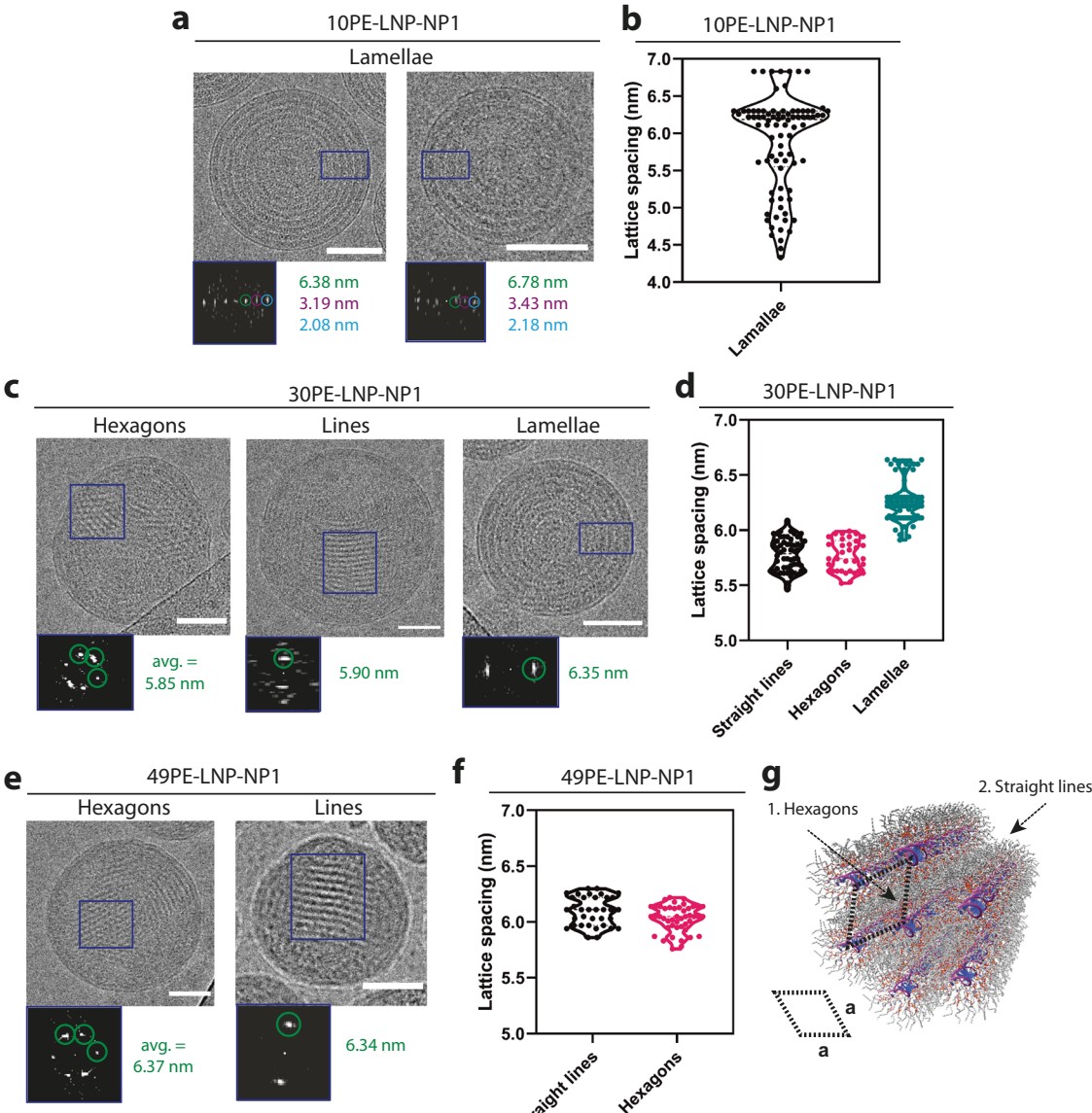

**Fig. 2 | Identification and quantification of lipid-siRNA structures in LNPs.**
**a** CryoTEM of representative structures found in 10PE-LNP-NP1. FFT values represent the [100] structure and higher-order reflections. Scale bars are 50 nm. **b** Quantification of lamellar spacings ($n = 84$, 5.88 nm ± 0.65 nm, median = 6.18 nm). Values are derived from the first-order reflections in the Fourier transform of 10PE-LNP-NP1. **c** CryoTEM of individual particles displayed representative structures found in 30PE-LNP-NP1. Scale bars are 50 nm. **d** Quantification of lattice spacings of the different structures displayed in **c**: straight lines ($n = 56$, 5.78 nm ± 0.14 nm), hexagons ($n = 34$, 5.77 nm ± 0.14 nm) and lamellae ($n = 78$, 6.26 nm ± 0.18 nm). **e** CryoTEM of individual particles displaying representative structures found in 49PE-LNP-NP1. Scale bars are 50 nm. **f** Quantification of lattice spacings of the different structures displayed in **e**: hexagons ($n = 46$, 6.03 nm ± 0.12 nm) and straight lines ($n = 35$, 6.10 nm ± 0.13 nm). **g** Atomistic model of inverse hexagonal lipid structures with associated primitive cell. The lattice spacings derived from cryo-TEM images are denoted as (**a**). Abbreviations used: NP itrogen to phosphate ratio, avg. average. Source data are provided as a Source Data file.

However, the median lattice spacing of 6.18 nm predominates, presumably causing the narrow peak observed by SAXS. Furthermore, this value closely reflects the expected values of lamellar lipid bilayers containing (phospho)lipids with similar carbon tail lengths ($C_{18}$) and cholesterol complexing oligonucleotides[48,49].

For 30PE-LNP-NP1 samples, we identified three distinguishable and repetitive structures: hexagons, straight lines and lamellae (Fig. 2c). Lattice spacing analysis of the individual structures in 30PE-LNP-NP1 revealed a high similarity for the straight lines and hexagons, at 5.78 nm ± 0.14 nm and 5.77 nm ± 0.14 nm respectively (Fig. 2d). These values are slightly smaller than the mean lattice spacings of $a = 6.26$ nm ± 0.18 nm observed for 10PE-LNP-NP1. Interestingly, LNPs with a hexagonal and lamellar structure were found to co-exist, and in some cases, these structures were visualized within the same particle

(Supplementary Fig. 8). This coexistence of LNPs with different structures and LNPs displaying coexisting structures within the same particle yields a broad scattering signal in the 30PE-LNP-NP1 SAXS profile instead of well-defined, distinguishable Bragg reflections (Fig. 1e). For 49PE-LNP-NP1, hexagonal and straight-line structures similar to 30PE-LNP-NP1 were observed (Fig. 2e). However, the presence of concentric lamellar structures was no longer observed. This absence of lamellar structures can be attributed to our rational design, given the high amount of DOPE lipids that possess a negative intrinsic curvature in the 49PE-LNP formulations. Quantification of the structures in 49PE-LNP-NP1 also yielded a high similarity in lattice spacing for straight lines and hexagons, at 6.10 nm ± 0.13 nm and 6.03 nm ± 0.12 nm respectively (Fig. 2f). This is in line with the presence of a very pronounced Bragg peak in the 49PE-LNP-NP1 SAXS pattern.

Altogether, the observation of straight lines and hexagonal structures in 30PE-LNP-NP1 and 49PE-LNP-NP1 clearly indicates the presence of an inverse hexagonal tubular structure, imaged from different viewpoints by cryoTEM (Fig. 2g). The expansion of the lattice of the inverted hexagonal phase from a = 5.78 nm ± 0.14 nm for 30PE-LNP-NP1 to a = 6.06 nm ± 0.13 nm for 49PE-LNP-NP1, can be explained by the increase in $C_{18}$-tailed DOPE and concomitant reduction in cholesterol content[50].

## Cryo-electron tomography reveals the formation of inverse spherical and liquid crystalline inverse hexagonal phases

Following the identification and characterization of liquid crystalline inverse hexagonal structures inside of LNPs using combined SAXS and cryoTEM, we sought to investigate the effect of siRNA content on the assembly pathway and three-dimensional architecture of the LNPs. Assembly of 49PE-LNPs without siRNA (49PE-LNP-noRNA) or with an NP ratio of 6 (49PE-LNP-NP6) both yielded straight lines and hexagonal structures similar to 49PE-LNP-NP1, but with different dimensions (Fig. 3a, Supplementary Fig. 7). Single particle lattice spacing analysis on these structures revealed a single population for 49PE-LNP-noRNA with an average lattice spacing of 4.78 nm ± 0.11 nm, and a bimodal distribution for 49PE-LNP-NP6 with an average of 4.84 nm ± 0.16 nm for the lower population and an average of 6.20 nm ± 0.17 nm for the

higher population (Fig. 3b). The lattice spacing of the lower population of the 49PE-LNP-NP6 is highly similar to the 49PE-LNP-noRNA distribution, whereas the lattice spacing of the higher population is highly similar to that of 49PE-LNP-NP1. This widening of the lattice spacing for the higher distribution of 49PE-LNP-NP6 and 49PE-LNP-NP1 over that of the lower population of 49PE-LNP-NP6 and 49PE-LNP-noRNA is due to encapsulation of siRNA within in the tubular structure, with the change being close to the calculated width ( ~ 2 nm) of a free siRNA molecule (Supplementary Fig. 1b). These therefore represent 'empty' and 'filled' hexagonally-packed tubular structures (Fig. 3c). Furthermore, no combination of 'empty' and 'filled' structures within the same LNP was observed with cryoTEM, suggesting that a non-saturating amount of siRNA during assembly gives rise to two parallel assembly pathways. DLS measurements indicated that 49PE-LNPs assembled without siRNA are larger and more polydisperse in dimension than LNPs assembled with a NP ratio of 1 (Supplementary Table 2). To examine this more closely, we compared the particle diameters of 'empty' and 'filled' 49PE-LNP variants determined by cryoTEM (Supplementary Fig. 9). In line with the DLS results, we found that 'filled' LNPs tend to be smaller and more monodisperse (D = 117 nm ± 33 nm and D = 118 nm ± 22 nm for 49PE-LNP-NP6-filled and 49PE-LNP-NP1, respectively) than 'empty' particles (D = 224 nm ± 70 nm and D = 257 nm ± 95 nm for 49PE-LNP-NP6-empty and 49PE-LNP-noRNA,

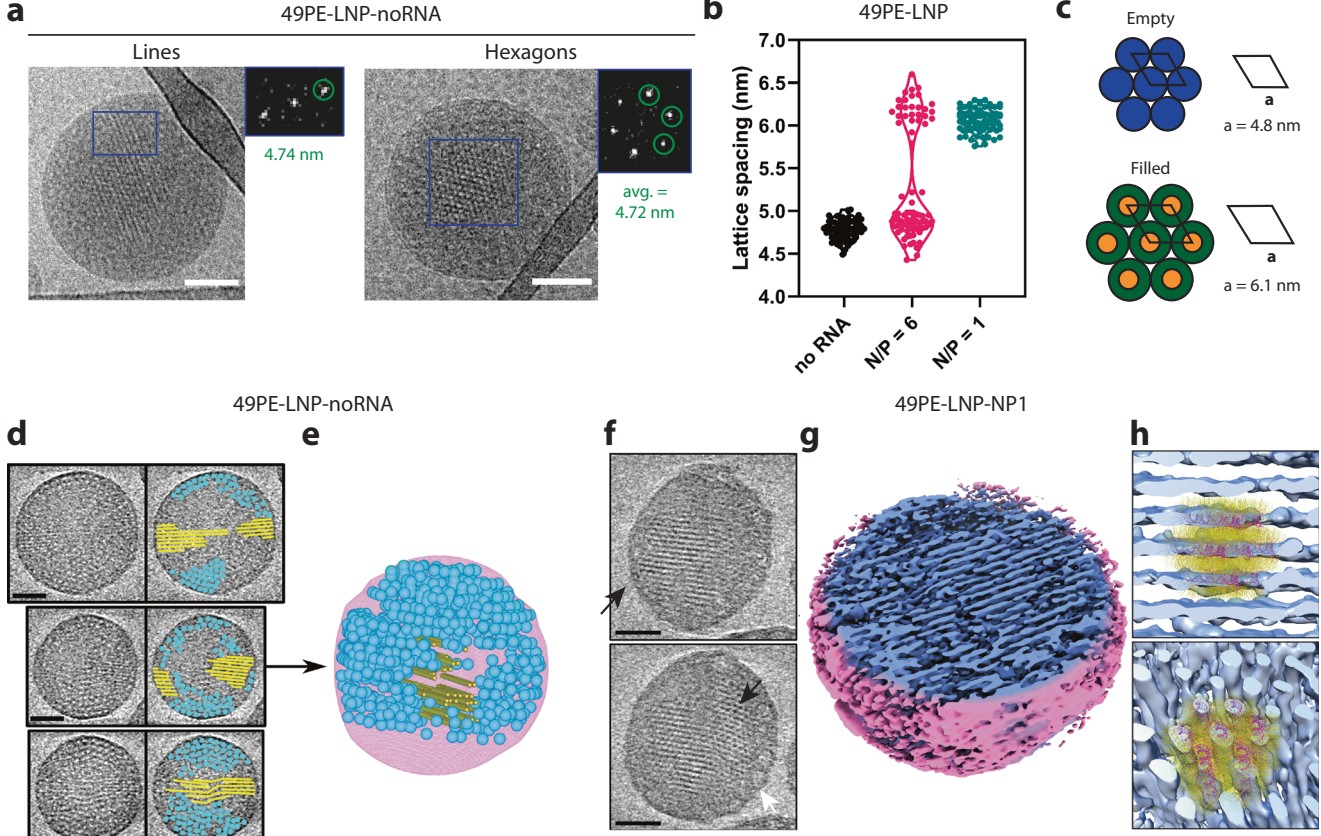

**Fig. 3 | Identification of filled and empty inverse hexagonal structures. a** Representative cryoTEM and FFTs of 49PE-LNP-noRNA showing two distinct structures: straight lines and hexagons. Scale bars are 50 nm. **b** Quantification of lattice spacings of structures found in 49PE-LNP at different RNA amounts: no RNA (n = 63, 4.78 nm ± 0.11 nm), NP = 6 (total: n = 77, 5.31 nm ± 0.67, low population: n = 50, 4.84 nm ± 0.16 nm, high population: n = 27, 6.20 nm ± 0.17 nm) and NP = 1 (n = 81, 6.06 nm ± 0.13 nm). **c** Schematic representation of 'empty' and 'filled' inverted hexagonal structures. **d** CryoET slices of 3 separate 49PE-LNP-noRNA particles. Identified spherical and tubular structures in each slice are indicated as blue circles and yellow lines respectively. Scale bars are 50 nm. Tomogram is a

representative selection of triplicate of reconstructions. **e** Reconstruction of the three-dimensional tomogram shown as the middle LNP in (**d**). **f** CryoET slices through an individual 49PE-LNP-NP1 particle. EM images represent slices at two different heights of the tomogram, showing straight lines (top) and hexagonally packed regions (bottom). Scale bars are 50 nm. **g** Surface rendering indicates internal liquid crystalline structures. **h** Fitted siRNA-lipid models to the electron density derived from cryoET, showing different orientations of the same structure (RNA in purple, lipid tails in yellow). Abbreviations used: NP = Nitrogen to phosphate ratio. Source data are provided as a Source Data file.

respectively). In addition, 49PE-LNP-noRNA shows the formation of empty tubular structures throughout the LNP core that were longer than the filled structures in the case of 49PE-LNP-NP1 (Supplementary Fig. 7b,d). Although the exact mechanisms underlying LNP assembly remain elusive and beyond the scope of this work[15,47,51], we hypothesize that DODAP and DOPE can form inverted hexagonal phases in the absence of siRNA, leading to the formation longer tubular structures which in turn favors the formation of larger LNPs. In the presence of siRNA, which is ~5 nm in length, the formation of long tubular structures requires the alignment of multiple siRNA molecules that, along with the electrostatic interaction between DODAP and siRNA, make it more prone to disruption and leads to the formation of smaller LNPs with liquid crystalline order comprising shorter tubes.

In addition to the inverse hexagonal tubular structures formed in 49PE-LNP-noRNA, we also identified a spherical structure that was able to co-exist in the same particle with inverse hexagonal tubular structures (Supplementary Fig. 10a). CryoET revealed the formation of two independent structures within the same LNP, and showed that the spheres resided at the periphery, in regions without inverse hexagonal tubes (Fig. 3d,e). In addition, 10 nm thick slices through the cryoET volume of an individual LNP showed that these spherical structures with a size of ~8-10 nm can be hexagonally packed (Supplementary Fig. 11). These spheres are reminiscent of an inverse micellar phase with an aqueous core[52]. In our case, inverse micelles consisting of DODAP and DOPE with the hydrophobic tails point outwards surrounding a large aqueous core accurately reflect the formed structure. The spheres and tubes in Fig. 3d,e therefore represent inverse micelles and empty hexagonally-packed tubular structures, respectively. In order to determine to what end the SAXS profiles reflected the formed phases we calculated the d-spacing using of the $q_{max}$ values for each of the distinguishable peaks. Due to the co-existence of phases, the first and second-order Bragg peaks could reflect [1,0] and [1,1] of one of the structures, or the [1,0] and [1,0] of the two individual hexagonally packed structures (spheres or tubes) (Supplementary Fig. 10b). Assuming a single structure, an average d-spacing of 8.48 nm was found, closely reflecting the average size of the inverse micellar structures identified in cryoET (Supplementary Fig. 10c,d), whilst calculations assuming two structures revealed that the d-spacing of the second order peak was ~4.95 nm, similar to the average lattice spacing of the inverse hexagonal tubular structures derived from cryoTEM (Fig. 3b). Although the resolution of SAXS may be insufficient for the accurate determination of multiple coexisting structural states, it still serves as a qualitative bulk measurement for the relative comparison of structural changes between LNP formulations. Furthermore, the increased lattice spacing in the case of 49PE-LNP-NP1 compared to 49PE-LNP-noRNA is confirmed by a shift in the SAXS profile to lower $q_{max}$ values, of which the same analysis yields and average d-spacing of 6.22 nm (Supplementary Fig. 10e,f). A similar shift of the first order peak was observed, suggesting the presence of hexagonally packed spheres with an average d-spacing of 10.71 nm in the case of 49PE-LNP-NP1. However, inverse micellar structures as found in 49PE-LNP-noRNA were not identified in the case of 49PE-LNP-NP1 using cryoTEM. Nevertheless, we did identify polymorphic-like structures that appear as disordered in cryoTEM (Supplementary Fig. 10g,h). Since the particles of 49PE-LNP-NP1 are significantly smaller than 49PE-LNP-noRNA, the identification of ~10 nm structures might be problematic in cryoTEM due to the lack of space at the LNP periphery, leading to a decreased repeatability and less order. To explore the structural organization in 49PE-LNP-NP1, we used cryoET to identify polymorphic and inverse hexagonal structures within a single LNP (Fig. 3f). Tomographic slices, 10 nm thick, through the volume revealed the presence of inverse hexagonally packed structures throughout the core of the LNP, with a spacing of 6.2 nm, equivalent to those observed by cryoTEM. The amorphous structures appeared on the LNP periphery, similar to the spherical structures in 49PE-LNP-noRNA, where they

form an interface between the well-ordered core and amorphous lipid membrane. Rendering the tomogram as an isosurface reveals the liquid crystalline nature of the inverse hexagonal phases more clearly, appearing in both straight lines and hexagonal orientations within the same particle (Fig. 3g), into which a model of a hexagonally-packed lipid-siRNA-lipid structure (50:50 mol% DODAP:DOPE) could be placed (Fig. 3h). The map shows a clear overlap of the maximum density with the siRNA molecules in regions displaying either straight lines and hexagonal orientations (Fig. 3h). Finally, since DOPE lipids are also able to form cubic phases in combination with specific lipids[53–55], the two obtained $q_{max}$ values were also calculated with the respective n-values for a cubic structure and that the d-spacings obtained here are not closely related and also do not reflect the d-spacings that were found in the cryoTEM analysis (Supplementary Table 4). These results indicate that the peaks in the SAXS profiles represent an overlap of two coexisting inverted hexagonally packed structures (tubes and micelles), but without the presence of cubic phases. Altogether, these results confirm the successful assembly of inverse hexagonal phases, encapsulating siRNA, in the core of LNPs.

## Liquid crystalline inverted structures are thermostable

The formation and characterization of LNPs is typically performed and reported at room temperature (RT, 25 °C). However, once applied at a physiological temperature (37 °C), the change in temperature can influence lipid structures and LNP performance. Therefore, we explored the stability of the lipid superstructures in 10PE-LNP-NP1, 30PE-LNP-NP1 and 49PE-LNP-NP1 at these temperatures using SAXS. Each LNP sample was subjected to three 6 h measurements and two 1 h temperature ramps to collect data at 25 °C before heating to 37 °C, and once again at 25 °C after heating. Interestingly, the Bragg reflection characteristic for the lamellar structure in 10PE-LNP-NP1 was only visible at room temperature before heating, while it was completely abolished at 37 °C and did not recover upon cooling down back to room temperature (Fig. 4a). In addition, no other Bragg reflections were observed. The loss of the lamellar structure and its transformation into a more disordered phase thus appears irreversible. By contrast, neither 30PE-LNP-NP1 nor 49PE-LNP-NP1 exhibited a significant change in their SAXS profiles collected at 25 °C and 37 °C (Fig. 4b,c). As a control formulation, we designed a LNP that would maintain its lamellar structure upon heating to 37 °C. We hypothesized that substitution of the ionizable lipid DODAP in 10PE-LNP-NP1 with its fully saturated analogue 1,2-distearoyl-3-dimethylammonium-propane (DSDAP), while keeping the rest of the lipids and NP ratio constant, would increase the overall transition temperature of the LNP formulation and lead to the formation of rigid lamellar structures (10PE-DS-LNP-NP1, Fig. 4d,e). This formulation showed a similar scattering signal compared to 10PE-LNP-NP1 at 25 °C, but no significant decrease when measured at 37 °C or after heating at 25 °C (Fig. 4f). We probed the dynamic structural changes in 10PE-LNP-NP1 and 10PE-DS-LNP-NP1 by collecting SAXS profiles at 25 °C, followed by measurements at 37 °C in intervals of 1 hour over a total time of 12 hours (Fig. 4g,h and Supplementary Fig. 12). For 10PE-LNP-NP1, the scattering signal decreased rapidly after 1 hour and was largely absent after 3-4 hours, whereas the scattering signal for 10PE-DS-LNP-NP1 remained generally unchanged over the complete 12-hour period. To confirm and visualize the structural changes in 10PE-LNP-NP1, cryoTEM imaging was performed after incubation for 1 hour at 37 °C, revealing the formation of polymorphic and amorphous structures, with a small amount of lamellar structure still present (Fig. 4i). For 10PE-DS-LNP-NP1, cryoTEM imaging showed the formation of spherical and disk-like LNPs containing lamellar structures at 25 °C (Fig. 4j). Notably, lamellar structures were not present as concentrical rings that extended to the core of the LNP, as found in 10PE-LNP-NP1, but instead showed the formation of amorphous structure in the LNP core. This is likely due to the severe decrease in bending capability of lamellar structures containing

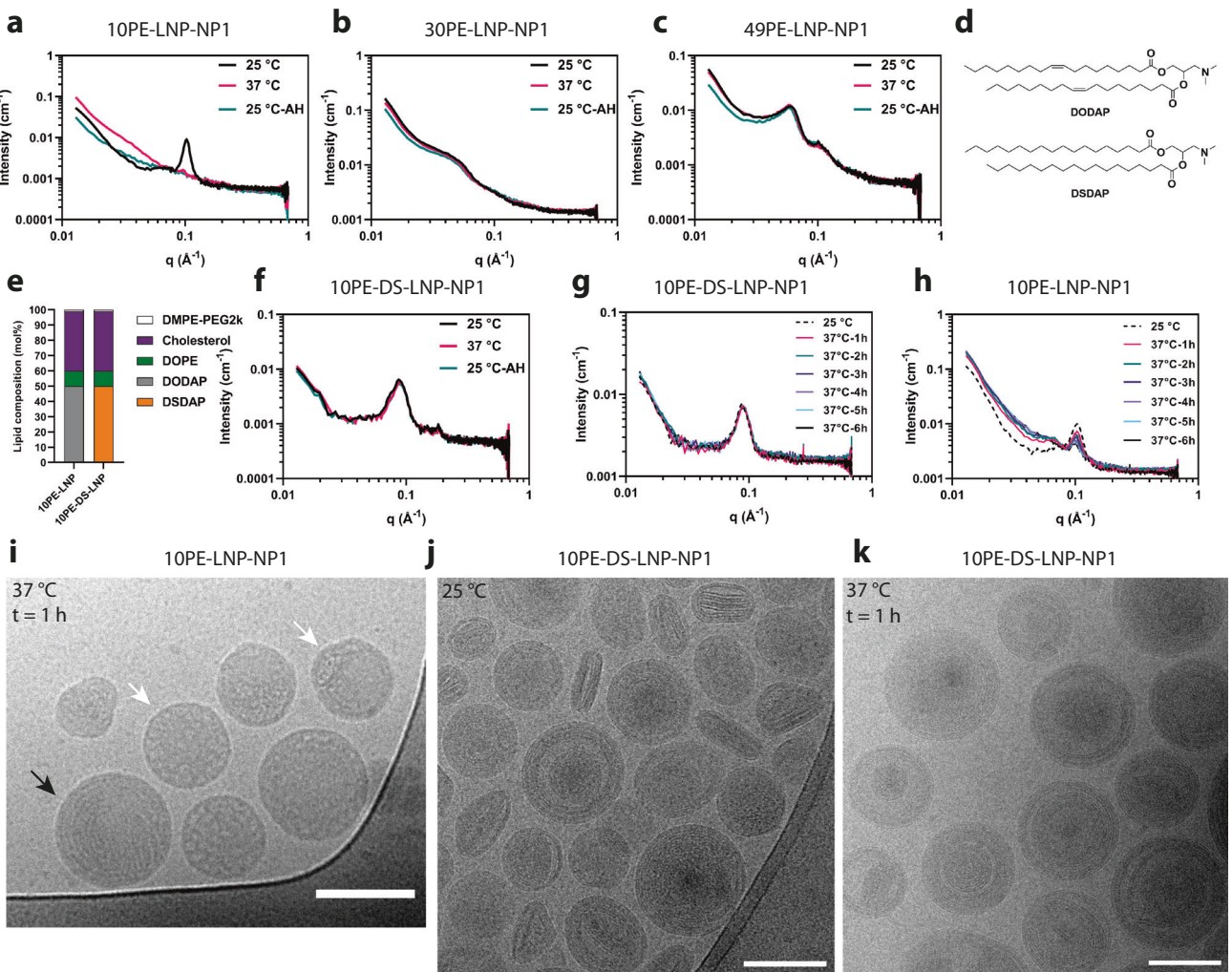

**Fig. 4 | The effect of temperature on LNP core structure. a–c** SAXS profiles of LNP-10PE-NP1, LNP-30PE-NP1 and LNP-49PE-NP1 at 25 °C, 37 °C and 25 °C after heating (AH). **d** Chemical structures of the ionizable lipids DODAP and DSDAP. **e** Lipid compositions for the rigidification of thermolabile lamellar 10PE-LNP-NP1 to a rigid formulation 10PE-DS-LNP-NP1. **f** SAXS profiles of 10PE-DS-LNP-NP1 at 25 °C, 37 °C and 25 °C after heating (AH). **g, h** SAXS profiles 10PE-DS-LNP-NP1 and 10PE-LNP-NP1 at 25 °C, and at 1 hour intervals at 37 °C. **i** CryoTEM of 10PE-LNP-NP1 after

incubation at 37 °C for 1 hour. White arrows indicate the formation of polymorphic and amorphous structures, black arrows indicate lamellar structure. Scale bar is 100 nm. **j, k** CryoTEM of 10PE-DS-LNP-NP1 at 25 °C and after incubation at 37 °C for 1 hour. Scale bars are 100 nm. Micrographs are a representative selection of a single experiment. Abbreviations used: NP = Nitrogen to phosphate ratio, h = hour. Source data are provided as a Source Data file.

the saturated and rigid DSDAP lipid. In addition, the observed structures of 10PE-DS-LNP-NP1 remained visible after a 1-hour incubation at 37 °C (Fig. 4k), in agreement with the respective SAXS profiles.

**Liquid crystalline inverse hexagonal phases in LNPs enhance silencing efficiency**

After characterization of LNPs, we assessed the intracellular silencing efficiency of LNPs containing rigid lamellar (10PE-DS-LNP-NP1), thermolabile lamellar (10PE-LNP-NP1), mixed (30PE-LNP-NP1) or liquid crystalline inverted hexagonal phases (49PE-LNP-NP1). The efficiency of LNP-mediated transfection was determined by the silencing of GFP tagged to the endogenous proteins CD63 and UBE21, in HeLa and U2OS cells, respectively[56,57]. In order to monitor the difference in uptake between formulations, a small amount of the non-exchangeable lipid dye 1,1′-dioctadecyl-3,3,3′,3′- tetra-methylindodicarbocyanine (DiD, 0.1 mol%) was added to the formulations during formation. The introduction of DiD into 49PE-LNP-NP1 still led to the formation of hexagonal structures with similar d-spacings as those without (Supplementary Fig. 13). These LNPs, encapsulating the siRNA for GFP, were added to the cells at various

doses of encapsulated siRNA (0 – 83 nM) and incubated for 24 hours, after which the medium was refreshed and the cells were allowed to grow for another 48 hours in order to deplete the GFP that was expressed before the addition of LNPs (Fig. 5a). At 72 hours, the cells were collected and fluorescence-activated cell sorting (FACS) was performed to quantify both the absolute uptake of LNPs, based on the mean fluorescence intensity (MFI) of the DiD dye, and the silencing efficiency relative to PBS treated cells from the MFI of GFP. A colorimetric cell-viability study (MTT) performed at 72 hours showed that the viability of both cultured cell lines remained unaffected by any of the LNP formulations across all doses (Supplementary Fig. 14).

For the U2OS-UBE21-GFP cell line, uptake of LNPs was similar in all cases up to 50 nM (Fig. 5a), but clear differences in GFP fluorescence were observed (Fig. 5b). By deriving the $IC_{50}$-values from the dose-response curves, we can clearly show that 49PE-LNP-NP1 ($IC_{50}$ = 9.9 nM) has a significantly increased transfection efficiency over 30PE-LNP-NP1 and 10PE-LNP-NP1 ($IC_{50}$ = 22.8 mM and 33.0 nM respectively) (Supplementary Table 3). Finally, the rigid lamellar LNP-10PE-DS-NP1 showed poor transfection ($IC_{50}$ = 55.7 nM) relative to the other formulations. For the latter, this observation of increased

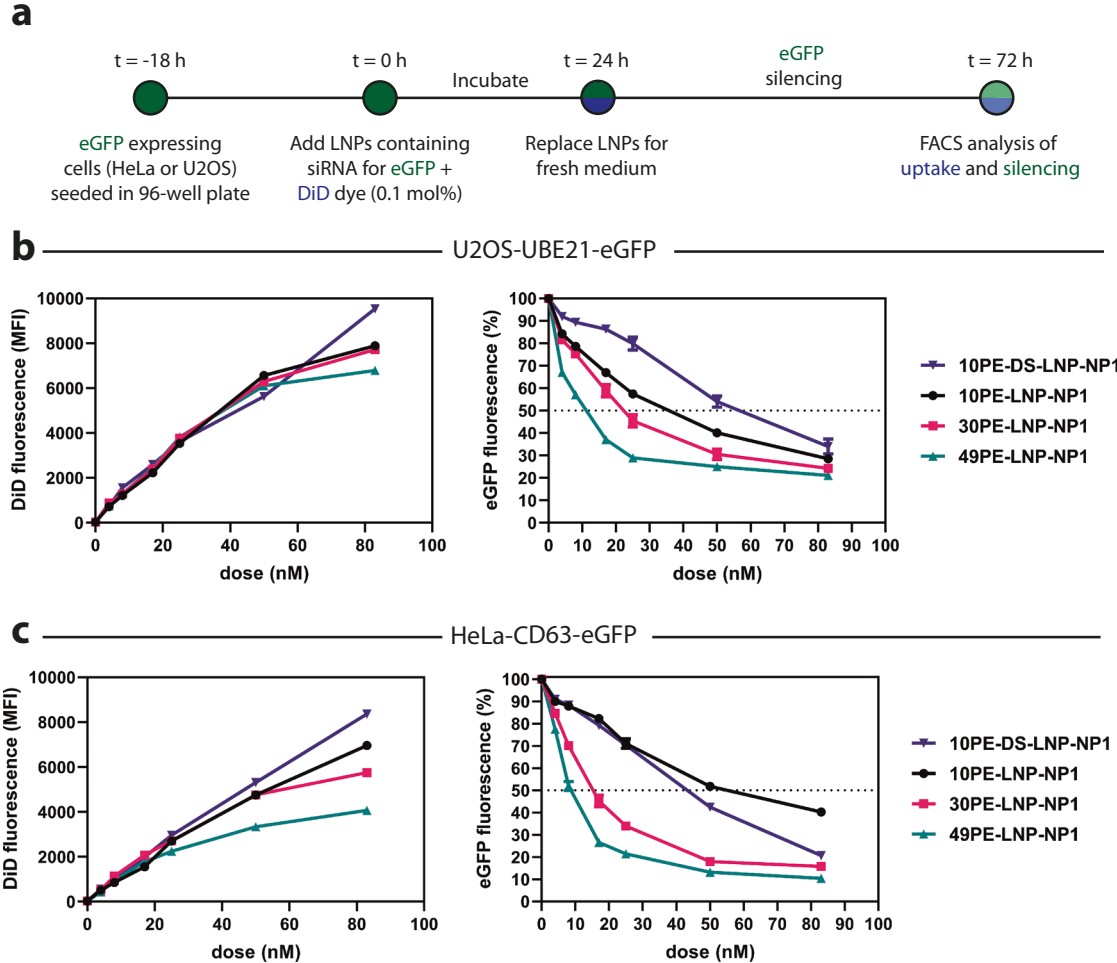

**Fig. 5 | Intracellular silencing of eGFP with lamellar and liquid crystalline inverted hexagonal LNPs. a** Schematic representation of the cell culture eGFP silencing experiment. Cells are incubated with LNPs for 24 hours, followed by medium replacement and culturing for an additional 48 hours. At 72 hours, the cells are collected and analyzed using FACS for LNP uptake and eGFP silencing. **b**, **c** LNP uptake and eGFP silencing in U2OS-UBE21-eGFP or HeLa-CD63-eGFP cell lines. LNP uptake is quantified based on the mean fluorescence intensity (MFI) of DiD in the cells. The relative eGFP fluorescence (%) is quantified as the MFI of GFP in the treated cells compared to PBS treated cells. All data points are the average of a triplicate and error bars reflect the standard deviation. Abbreviations used: NP Nitrogen to phosphate ratio, h hour. Source data are provided as a Source Data file.

saturation in ionizable cationic lipids leading to less efficient cellular transfection closely reflects previous reports[58]. In the HeLa-CD63-eGFP cell line, similar uptake and silencing efficiency trends were observed for all formulations up to 25 nM (Fig. 5c, Supplementary Table 3). Similar trends of silencing efficiency were observed to the U2OS cell line, in which 49PE-LNP-NP1 ($IC_{50} = 9.0$ nM) and 30PE-LNP-NP1 ($IC_{50} = 15.3$ nM) performed the best, despite lower cellular uptake at higher doses (50 nM and 83 nM). However, in this cell line, 10PE-LNP-NP1 ($IC_{50} = 57.2$ nM) and 10PE-DS-LNP-NP1 ($IC_{50} = 39.7$ nM) exhibited lower but differential silencing efficiencies suggesting that there is a cell-dependent behavior regarding the rigid and thermo-labile lamellar structures. A comparison of 49PE-LNP-NP1 ± 0.1 mol% DiD also showed that the transfection efficiency was also not affected by the introduction of the fluorescent dye into the LNPs (Supplementary Fig. 15). In addition, 49PE-LNP-NP1 encapsulating a negative control siRNA showed that there was no LNP-induced background transfection occurring (Supplementary Table 1, Supplementary Fig. 15).

**Pre-programmed inverse hexagonal structures remain stable upon interaction with LUVs**
After establishing a clear trend in transfection efficiency when comparing LNPs with defined lipid structures, we sought to gain more mechanistic insights into the LNP-membrane interaction. Interactions

between LNPs and endosomal membranes are driven by an electrostatic interaction of negatively charged membranes and protonated LNPs in an acidic endosomal environment[17]. Therefore, we studied the interaction of LNPs with anionic large unilamellar vesicles (LUVs) that mimic endosomal membranes. The lipid composition of endosomal membranes is complex, dynamic and dependent on cell type, typically rich in (zwitterionic) neutral lipids such as phosphatidylcholine (PC), phosphatidylethanolamine (PE) and cholesterol (Chol)[59], along with lipids that have a formal negative charge such as phosphatidylserine (PS) and phosphatidylinositol (PI)[60,61]. A lipid composition of PC:PE:PS:Chol:PI at a ratio of 50:27:10:10:3 mol% was chosen, where purified lipid extracts with variable lipid chain lengths for each of the phospholipids were used, to reflect the high variability of lipid chain length and saturation in natural membranes (see Supplementary Information). These unilamellar model membranes were assembled using thin-film hydration, followed by multiple freeze-thawing cycles and extrusion. CryoTEM, DLS and Zeta Potential measurements confirmed the formation of monodisperse anionic LUVs of ~100–150 nm (Supplementary Fig. 16, Supplementary Table 2).

We first sought to investigate the effect that the different LNP lipid compositions had on lipid mixing with our model membranes. To this end, we added two DOPE lipid-conjugated fluorophores, Nitrobenzoxadiazole (PE-NBD) and Lissamine Rhodamine B (PE-LR) to the

composition of the LUVs at 1.5 mol% each, substituting for PE. These fluorophores form a Fluorescence Resonance Energy Transfer (FRET) pair, in which PE-LR quenches the fluorescence of PE-NBD when they are in close proximity. Upon lipid mixing between LNP and LUVs, the lipid-conjugated fluorophores will be separated leading to dequenching of the PE-NBD fluorescence over time (Fig. 6a). When lipid mixing was performed at a typical endosomal pH of 6.0 at 37 °C[62], all LNPs displayed lipid mixing with the acceptor liposomes, but a clear trend was observed showing that an increase in DOPE content displayed faster lipid mixing kinetics as well as a higher total lipid mixing efficiency (Fig. 6b). Furthermore, the rigid lamellar formulation LNP-10PE-DS-NP1 showed the lowest lipid mixing kinetics and efficiency. In all cases, no active lipid mixing was observed at a physiological pH of 7.4 at 37 °C, confirming the need for a pH below the pKa of DODAP for the initiation of the LNP-LUV interaction (Fig. 6b). The trend of lipid mixing efficiency closely correlates to the observed silencing efficiencies observed in cell culture (Fig. 4b,c), where 49PE-LNP-NP1 is the most efficient for both lipid mixing and silencing.

Next, we explored the alteration of the defined lipid structures upon lipid mixing of LNPs with LUVs. Judging from the FRET-based assay, lipid mixing reached an equilibrium after ~30 minutes. Therefore, we visualized the LNP-LUV interaction with cryoTEM after 1 and 7 hours and used SAXS in the intermediate period of 6 hours to determine the preservation of defined lipid structures (Fig. 6c). The very inefficient lipid mixing of 10PE-DS-LNP-NP1 was reflected in the SAXS profile, showing a minor decrease in lamellar signal compared to 10PE-DS-LNP-NP1 before mixing (Supplementary Fig. 17a,b). In addition, cryoTEM revealed the docking of LNPs to LUVs but lacking further fusion or structural change, at both 1 and 7 hours. (Supplementary Fig. 17c,d).

In the case of 10PE-LNP-NP1, the respective SAXS profile showed a decrease in the lamellar signal after mixing (Fig. 6d). The SAXS profile also shows the increase of a distinguishable peak at a lower q-value, suggesting the formation of additional structures. CryoTEM revealed that these LNPs interact with LUVs and form inverted hexagonal structures at the LNP-LUV interface, while also retaining characteristics of lamellar structures as determined by FFT analysis (Fig. 6f, Supplementary Fig. 17). Furthermore, these structures remain visible after incubation with LUVs after 7 hours, indicating preservation of the lipid structures. The first-order peaks of both lamellar and inverse hexagonal structures will appear as a signal at a q-value of ~0.1, making it difficult to distinguish the contribution of these structures to the observed signal. Nevertheless, a complete abolishment of the lamellar structures as seen for 10PE-LNP-NP1 alone is not observed, indicating that the thermal stability of these structures is increased upon lipid mixing with the acceptor LUVs.

For 30PE-LNP-NP1 mixed with LUVs, the measured SAXS profile showed complete abolishment of the broad 30PE-LNP-NP1 signal present before mixing while primarily yielding a signal at a q-value of ~0.1 (Supplementary Fig. 18a,b). CryoTEM at both 1 and 7 hours show the primary formation of inverted hexagonal structures and straight lines, although some lamellar structures were also identified (Supplementary Fig. 18c,d). Finally, for 49PE-LNP-NP1, the SAXS profile showed the complete abolishment of the signal corresponding to the first-order signal of the larger structures, yielding a clearly defined peak corresponding to the first-order peak of the inverse hexagonal tubular structures (Fig. 6e, Supplementary Fig. 10e). CryoTEM showed the sole formation of crystalline inverse hexagonal phases, visualized as hexagons or straight lines (Fig. 6g, Supplementary Fig. 20). Furthermore, these structures remained clearly visible after 7 hours, confirming preservation upon interaction with the LUVs and the thermal stability of these structures.

From these results, we propose a mechanism that explains the observations of enhanced lipid mixing and transfection of LNPs with liquid crystalline inverse hexagonal over lamellar phases, and to what extent it might impact intracellular siRNA delivery (Fig. 6h). After an initial interaction of LNPs with acceptor LUVs triggered by the protonation of ionizable lipids (ILs) at acidic pH, the LNPs follow two different pathways for the delivery of their cargo. In the case of 10PE-LNP-NP1, an in situ transition of lamellar to inverse hexagonal phase takes place at the LNP-LUV intersection, which allows for membrane disruption and the eventual escape of RNA molecules from the LNP (Fig. 6h, pathway 1). These findings are complementary to the rationale for which membrane-disrupting ILs or LNP formulations were designed, in which ILs form non-lamellar structures with the anionic lipids of the acceptor membrane[18,19]. Other factors that will play a role in LNPs with lamellar structures can be the overall fluidity of the lipid mixture and the pKa of the IL[22,25]. Although the formation of pre-programmed inverse hexagonal phases in LNPs has not been reported before, permanently cationic lipid-oligonucleotide complexes, named lipoplexes, show a degree of similarity to our system[63–65]. Here, lipoplexes containing inverse hexagonal phases encapsulating double-stranded DNA (dsDNA) interacting with acceptor giant unilamellar vesicles (GUVs, ~20 μm) are shown to generate near complete lipid mixing towards a lamellar acceptor membrane[49]. However, the visualization of lipoplex-membrane complexes at the nanoscale has not been reported. More recently, molecular dynamics simulations of inverse hexagonal lipoplexes containing dsDNA interacting with anionic membranes show how dsDNA is able to escape from the lipoplex[66]. These results align directly with our observation of retention of the pre-programmed inverse hexagonal phase upon interaction with the LUVs. Derived from our cryoTEM and SAXS data, we similarly demonstrate that there is a lamellar-to-inverse hexagonal phase transition is occurring for 10PE-LNP-NP1, whereas the pre-programmed inverse hexagonal phases from the 49PE-LNP-NP1 remains intact upon interaction with the LUVs (Fig. 6h). An important observation is, although the presence of liquid crystalline $H_{II}$ phases improves lipid mixing and transfection efficiency, complete fusion of the LNP-LUV systems does not occur, leaving a large amount of siRNA complexed in thermodynamically stable siRNA-lipid structures. Although our in vitro model is a simplified version of the LNP-membrane interaction and lacks the presence of the complex dynamic intracellular environment, these observations and proposed mechanism may provide a contributing factor to the poor endosomal escape of RNA molecules (≤ 2%) from the endosome into the cytosol[17,20,21].

## Outlook

Using bottom-up rational design of defined lipid superstructures, we have generated LNPs with liquid crystalline inverse hexagonal lipid structures actively encapsulating siRNA and with enhanced transfection efficiency over LNPs containing lamellar structures. Due to the inherent complexity of lipid behavior in LNPs during and after nanoparticle assembly, an in-depth characterization combining cryoTEM, SAXS and cryoET was employed to determine the effects of lipid composition, RNA content, temperature and membrane interaction on the formation and stability these structures. Endosomal escape remains a tremendous barrier for the improvement of LNP-based RNA therapeutics, yet our data shows that, through inducing pre-programmed lipid phases, transfection efficiency can be enhanced. This may play a crucial role for widespread applications. Here, it is important to recognize the extensive efforts in the exploration of the chemical space of ionizable lipids[24,25,67], and other LNPs components for the promotion of endosomal escape of RNA molecules[33,34]. Since the formation of non-lamellar lipid structures is driven by total lipid composition of these multicomponent systems, and therefore not exclusively on specific ILs, we believe that our approach for inducing inverse hexagonal structures can be retrofitted to LNPs with other ILs or components, albeit it at different lipid concentrations or RNA content, to increase their LNP potency, which will allow additional exploration of the structural space of LNPs. Consequently, the large

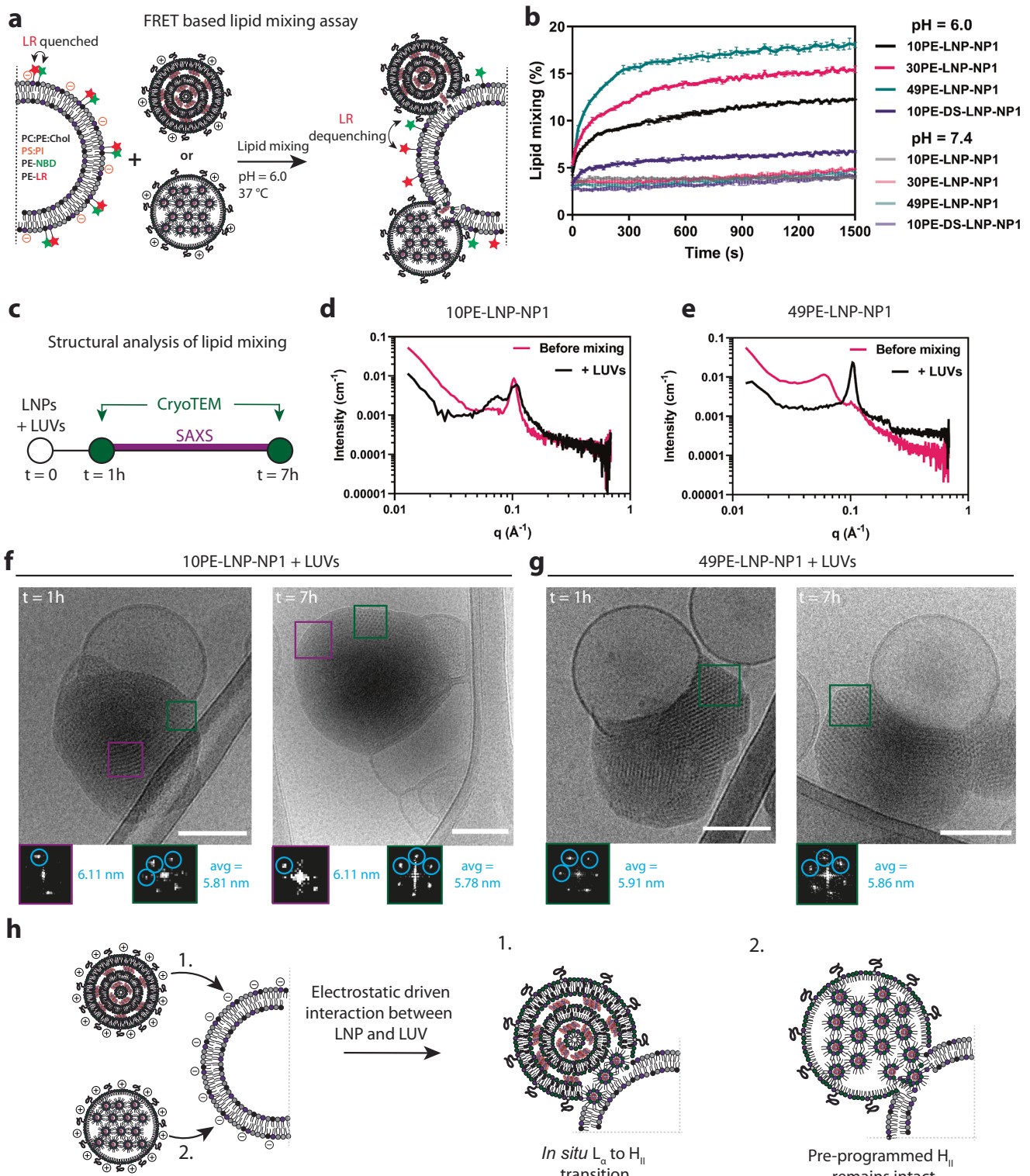

**Fig. 6 | Interaction of LNPs with anionic LUVs. a** Schematic representation of the FRET based lipid mixing assay. LNPs are mixed with anionic acceptor liposomes containing two lipid conjugated fluorophores (PE-NBD and PE-LR), after which the dequenching of PE-NBD is measured over time. **b** Lipid mixing results of LNPs (250 μM) mixed with anionic LUVs (125 μM) at pH 6.0 and 7.4 cryoTEM, performed at 37 °C for 25 minutes. Plotted data is the average of a triplicate in which error bars represent the standard deviation. **c** Schematic representation of the structural analysis of lipid mixing between LNPs and LUVs. CryoTEM imaging was performed after an incubation of 1 hour at 37 °C. Similarly, SAXS measurements over a time span of 6 hours were started after an incubation of 1 hour at 37 °C. **d, e** SAXS profiles of 10PE-LNP-NP1 or 49PE-LNP-NP1 mixed with LUVs. **f, g** Representative cryoTEM images of 10PE-LNP-NP1 or 49PE-LNP-NP1 mixed with LUVs. Micrographs are a representative selection from a duplicate of experiments. Scale bars are 100 nm. **h** Schematic representation of the interaction of LNPs with LUVs, depicting the change in situ transition from lamellar to inverted phases for 10PE-LNP-NP1, where the pre-programmed hexagonal phases in the 49PE-LNP-NP1 remains intact. Abbreviations used: NP Nitrogen to phosphate ratio, LUV large unilamellar vesicle, FRET Fluorescence resonance energy transfer. Source data are provided as a Source Data file.

variety of natural and synthetic lipids also generates possibilities for the rational design of other defined lipid structures[41,54,68].

LNPs are currently applied in a widespread of LNP-RNA therapeutic approaches[69–72], for example, in the form of prophylactic vaccines against COVID-19[11,12,73,74]. It will be of great interest to expand the defined structural space of LNPs containing RNA molecules with a more complex and less predictable structure than siRNA (e.g. mRNA and single guide RNA). To this end, bottom-up approaches of LNP design and application, based on rational design and the mechanistic understanding of the nano-bio interactions, can help to predict and understand of these structures. Finally, mechanistic knowledge about lipid superstructures containing RNA molecules, and their behavior with endosomal membranes, will enable the predictions of LNP potency from a bottom-up based approach in which LNP formulations are assessed with biophysical methods before their application to in vitro or in vivo systems, opening up another avenue of LNP design and optimization.

## Methods

### Reagents
Cholesterol was purchased from Sigma-Aldrich (Zwijndrecht, The Netherlands). 1,1′-Dioctadecyl-3,3,3′,3′-Tetramethylindodicarbo cyanine (DiD) was purchased from Thermo Fisher Scientific (Landsmeer, The Netherlands). L-α-phosphatidylserine (Brain, Porcine, PS), L-α-phosphatidylcholine (Brain, Porcine, PC), L-α-phosphatidylethanolamine (Brain, Porcine, PE), L-α-phosphatidylinositol (Brain, Porcine, PI), 1,2-dioleoyl-sn-glycero-3-phosphoethanolamine-N-(7-nitro-2-1,3-benzoxadiazol-4-yl) (PE-NBD), 1,2-dioleoyl-sn-glycero-3-phosphoethanolamine-N-(lissamine rhodamine B sulfonyl) (PE-LR), 1,2-dioleoyl-sn-glycero-3-phosphoethanolamine (DOPE), 1,2-dimyristoyl-sn-glycero-3-phosphoethanolamine-N-methoxypolyethylene glycol-2000 (DMPE-PEG2k), 1,2-dioleoyl-3-dimethylammonium-propane (DODAP) and 1,2-distearoyl-3-dimethyl-lammonium-propane (DSDAP) were purchased from Avanti Polar Lipids through Merck. All siRNA molecules were purchased from Integrated DNA Technologies (Leuven, Belgium) through custom synthesis, exact sequences can be found in Supplementary Table 1 and are provided in the source data. HeLa-CD63-eGFP, U2OS-UBE21-eGFP cell line were kindly provided by Dr. Ilana Berlin (LUMC, Leiden) and cultured in DMEM growth medium (Sigma Aldrich) containing sodium bicarbonate, without sodium pyruvate and HEPES, was supplemented with 10% fetal bovine serum (Sigma), 1% of L-glutamine (Thermo Fisher Scientific) and 1% penicillin/streptomycin (Thermo Fisher Scientific), at 37 °C in the presence of 5% CO2. Opti-MEM reduced serum medium (Thermo Fisher Scientific) was applied among transfection experiments. All cell lines were routinely tested (negatively) for mycoplasma.

### Self-assembly of lipid nanoparticles (LNPs)
Lipids were combined at the desired molar ratios and concentrations from stock solutions (1-10 mM) in chloroform:methanol (1:1). Solvents were evaporated under a nitrogen flow and residual solvent was removed *in vacuo* for at least 30 minutes. The lipid film was dissolved in absolute ethanol and used for the assembly. A solution of siRNA was made by dissolving siRNA in 50 mM citrate buffer (pH = 4, RNase free). The solutions were loaded into two separate syringes and connected to a T-junction microfluidic mixer. The solutions were mixed in a 3:1 flow ratio of siRNA against lipids (1.5 mL/min for siRNA solution, 0.5 mL/min for lipids solution). After mixing, the solution was directly loaded in a 10k MWCO dialysis cassette (Slide-A-Lyzer™, Thermo Scientific) or a 1 MDa MWCO dialysis cassette (Spectra-Por® Float-A-Lyzer® G2, Thermo Scientific) and dialyzed against Phosphate Buffered Saline (PBS, 137 mM NaCl, 2.7 mM KCl, 8 mM $Na_2HPO_4$, and 2 mM $KH_2PO_4$) overnight. After overnight dialysis, siRNA encapsulation efficiency was determined by Quant-iT™ RiboGreen™ RNA Assay Kit as described below. If necessary, LNPs were concentrated using 100k

MWCO centrifugal filters (Amicon® Ultra, Merck). Adjustment and dilution of LNPs was done with Dulbecco's PBS (Merck).

### RNA encapsulation and dose determination assay
Encapsulation efficiency (EE%) defined as the amount of siRNA encapsulated versus the free siRNA in solution after dialysis was determined using a Quant-iT™ RiboGreen™ RNA Assay Kit (Invitrogen). For the determination of non-encapsulated siRNA, LNPs after dialysis were diluted with the supplied 1 x TE buffer (RNase free) and treated with the RiboGreen™ reagent. For the determination of the complete amount of siRNA in the sample, LNPs after dialysis were treated with 1% Triton X-100 in TE buffer (RNase free) and incubated for 15 minutes followed by dilution with TE buffer and treatment with the Ribo-Green™ reagent. Both conditions were performed in triplicate to ensure proper lysis of the LNPs. Change in fluorescence was measured in 96-well plates using a TECAN Infinite M1000 Pro microplate reader and the percentage of siRNA encapsulation (EE%) was determined using the fraction of $(F_{total\ RNA} - F_{free\ RNA})/F_{total\ RNA} \times 100\%$. Quantification of the dose was determined using a similar protocol, in which the RiboGreen™ fluorescence was determined of concentrated LNPs. The supplied RNA standards were used to generate a standard curve and the concentration of encapsulated mRNA was determined by inserting $(F_{total\ RNA} - F_{free\ RNA})$ in the standard curve.

### Cryogenic transmission electron microscopy (cryoTEM)
Vitrification of concentrated LNPs (~10-15 mM) was performed using a Leica EM GP operating at 22 °C or 37 °C and 95% relative humidity. Sample suspensions were placed on glow discharged 150 μm lacey carbon films supported by 200 mesh copper grids (Electron Microscopy Sciences). In general, optimal results were achieved using a 30-60 second pre-blot and a 1 second blot time. After vitrification, sample grids were maintained below −170 °C and imaging was performed on a Tecnai T12 (ThermoFisher) with a biotwin lens and LaB6 filament operating at 120 keV equipped with an Eagle 4 K x 4 K CCD camera (ThermoFisher). Images were acquired at a nominal underfocus of −2 to −3 μm (49,000× magnification) with an electron dose of ~2000 e⁻·nm⁻².

For higher resolution cryoEM, grids were loaded into a Titan Krios transmission electron microscope (FEI Company) equipped with a field emission gun operating at 300 kV and were imaged using a Falcon 3 direct electron detector (FEI). Images were acquired a calibrated magnification of 75,000× with a nominal underfocus of −1 to −2 μm and an electron dose of ~2000 e⁻·nm⁻².

### Single particle analysis of cryoTEM data
Single particles were chosen based on the presence of structure formation from a collection of 2D cryoTEM images, divided over three independent assemblies of the same LNP formulation. Only particles present in vitreous ice were considered for analysis. Regions of interest from individual particles and the fast Fourier transform (FFT), as well as the determination of particle size was performed using the Fiji distribution of ImageJ[75].

### Cryogenic electron tomography (cryoET)
Samples for cryoET were prepared as described for 2D cryoTEM, but with the addition of BSA-coated 10 nm gold beads immediately prior to vitrification to act as fiducial markers for tomogram reconstruction. Tilt series were collected using Tomo 4.0 (FEI) on a Titan Krios at a magnification of 75,000×, for a final pixel size 1.87 Å/pixel, using a continuous tilt series from −60° to +60° with increments of 2° and an electron dose of ~200-250 e⁻/nm² per tilt image. Focusing to −4 to −5 μm was performed every second image acquisition using a low-dose routine. Tomograms were processed using IMOD software (v4.x)[76], using fiducial tracking and reconstructed with both weighted back-projection and 5 iterations of a SIRT-like filter within IMOD to enhance

contrast. Tomograms were visualised and analysed using a combination of IMOD and UCSF Chimera (v1.17)[77].

## Model building
DOPE and DODAP lipids were built using PyMOL (v2.x)[78], before parametrization in eLBOW within Phenix (v1.19)[79,80]. Lipids were relaxed into varied appropriate geometries using ISOLDE within UCSF ChimeraX (v1.17)[81,82]. UCSF Chimera (v1.17) was used to build models of siRNA and lipid-siRNA structures.

## Small Angle X-ray scattering (SAXS)
SAXS measurements were performed in transmission mode on a SAXSLAB GANESHA system with a Pilatus 300 K solid-state photon-counting 2D detector using a high brilliance Microfocus Cu Source, Xenocs Genix3D, wavelength 1.54184 Å. The LNPs ( ~ 12 mM) were loaded into 2 mm lockable thin wall capillaries and measured at a q-range of $0.0129 - 0.6870 Å^{-1}$ with an exposure time of 6 hours. Prior to each measurement series, a silver behenate standard was used to correct for deviations in the sample to detector distance. For temperature-dependent experiments, the temperature was equilibrated to 25 °C or 37 °C for 30 minutes prior to measurement. The measured SAXS profiles are displayed as the average intensity I(q) vs. the q-range. For kinetic temperature experiments, the measurement time was divided into the average intensity of 1-hour sections.

## Assembly of large unilamellar vesicles (LUVs)
LUVs were assembled using a combination of freeze-thaw cycles and extrusion. Lipids were combined from stock solutions (2-10 mM) in chloroform to a achieve a lipid composition of PC:PE:PS:Chol:PI of 50:27:10:10:3 mol% or PC:PE:PS:Chol:PI:PE-NBD:PE-LR of 50:24:10:10:3:1.5:1.5 mol%. The solvent was evaporated from the mixture under a nitrogen flow, followed by removal of trace solvents *in vacuo* for at least 1 hour. The resulting lipid film was hydrated with either 100 mM citrate buffer (pH = 5.5) or phosphate buffered saline (PBS, pH = 7.3), to achieve the desired total lipid concentration (10-20 mM) and vortexed until the entire lipid film was fully suspended in solution. The suspension was subjected to seven freeze-thaw cycles. In each cycle, the mixture was frozen completely with liquid nitrogen and left to thaw at room temperature, followed by vigorous mixing to ensure complete thawing. After these cycles, the mixture was extruded at 40 °C (Mini-extruder, Avanti Polar Lipids, Alabaster, US). The mixture was passed 11 times through a 200 nm polycarbonate (PC) membrane (Nucleopore Track-Etch membranes, Whatman). All liposome formulations were stored at 4 °C and used within 2 days.

## Lipid mixing determined by Fluorescence Resonance Energy Transfer (FRET)
For the determination of lipid mixing efficiency by FRET, LUVs with a composition of PC:PE:PS:Chol:PI:PE-NBD:PE-LR of 50:24:10:10:3:1.5:1.5 mol% were prepared in 100 mM citrate buffer (pH = 5.5) or PBS (pH = 7.4) as described above. The LUVs were diluted with their respective buffers to 250 μM and 100 μL was transferred to black F-bottom chimney 96-well plates (Greiner®). The LUVs were heated to 37 °C prior to further use. For the control serving as 100% lipid mixing, 20 μL of 1% Triton X-100 in $H_2O$ was added followed by the addition of 80 μL PBS. For the control serving as 0% lipid mixing, 100 μL PBS was added to the LUVs. LNPs were assembled and diluted in PBS (pH = 7.4) to 500 μM and 100 μL was added to the acceptor LUVs, yielding a final concentration 125 μM LUVs and 250 μM LNPs. In the case where 100 mM citrate buffer was used for the LUVs, the final pH was ~6.0. All of the conditions were performed in triplicate. After addition of the LNPs, the dequenching of the PE-NBD signal was measured every 15 seconds for 25 minutes using a TECAN Infinite M1000 Pro microplate reader (λ: excitation = 460 nm ± 10 nm, emission = 535 nm ± 10 nm).

## CryoTEM and SAXS analysis of lipid mixing
For the assessment of lipid mixing with cryoTEM and SAXS analysis, LUVs with a composition of PC:PE:PS:Chol:PI of 50:27:10:10:3 mol% were prepared in 100 mM citrate buffer (pH = 5.5) at a concentration of 20 mM as described above. LNPs were prepared as described before and concentrated to ~20 mM. The LUVs and LNPs were mixed at a volumetric ratio of 1:2, respectively, and incubated at 37 °C. For cryo-TEM, samples were vitrified after 1 hour of incubation as described above. In the case of SAXS measurements, the mixture of LUVs and LNPs were loaded into capillaries and equilibrated at 37 °C for 1 hour prior to the start of the measurement, as described above. The presented data is of the mixed samples minus the background of LUVs in citrate buffer mixed with PBS (without LNPs) at the same final concentration.

## Cell transfection and FACS experiments
HeLa-CD63-eGFP, U2OS-UBE21-eGFP were seeded in 96-well plate at the density of 1*10⁴ cells/well on the day before, different concentrations of siRNA-GFP encapsulated LNPs (0 – 83 nM) in 100 μL Opti-MEM medium were added to the cells and incubated for 24 h, then the medium was removed and refreshed with new Opti-MEM for continuous 48 h culturing. After 72 h incubation, cells were digested, collected and resuspended in PBS for FACS measurements (Guava easyCyte Flow Cytometers). DiD mean fluorescence intensities (Red-R channel) was quantified as the uptake of LNPs into cells. The eGFP expression (GFP MFI, Green-B channel) of all LNP treated cells are relative to PBS treated cells, for which the values are set as 100%.

## Cell viability assay
HeLa-CD63-eGFP, U2OS-UBE21-eGFP cells were seeded 96-well plate at a density of 1*10⁴ cells per well the day before, then followed with the same procedure as the transfection with different concentrations of LNPs determined by the Ribogreen RNA assay. After 72 h incubation, cell viability reagent alarmaBlue HS solution (10 μL, ThermoFisher) was added to the medium (100 μL) and incubated for another 4 h at 37 °C. After 4 h, the absorbance at 570 nm (using 600 nm as a reference wavelength) was measured at room temperature using a Tecan infinite M1000, which was shaken for 60 s before measurement (2 mm linearly, 654 rpm). The cell viability was normalized with control (blank HeLa-CD63-eGFP, U2OS-UBE21-eGFP cells), which was set at 100% cell survival. All conditions were performed in triplicate.

## Reporting summary
Further information on research design is available in the Nature Portfolio Reporting Summary linked to this article.

# Data availability
CryoTEM micrographs from all the figures in the main text and supplementary information have been deposited within the public image database figshare.com (10.6084/m9.figshare.24456781). Source data are provided with this paper.

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

## Acknowledgements

The authors thank Dr. Ilana Berlin for kindly gifting the HeLa-CD63-eGFP and U2OS-UBE21-eGFP cell lines. This work benefited from access to the Netherlands Centre for Electron Nanoscopy (NeCEN) at Leiden University, an Instruct-ERIC center with assistance from W.E.M Noteborn and L. Renault. The research was financially supported via a NWO VICI grant (724.014.001) to A.K and by a European Research Council grant (759517) to T.H.S.

## Author contributions

R.P. and T.H.S. conceived the research. R.P. designed and performed all experiments unless stated otherwise and analyzed the data. T.H.S. assisted in the cryoTEM analysis and performed the cryoET reconstructions. Y.Z. performed the cell culture and FACS experiments and analysis. M.M.R.M.H. performed the SAXS measurements. M.M.R.M.H. and I.K.V. assisted in the design and analysis of SAXS experiments. A.K. and T.H.S. supervised the research. R.P. and T.H.S. wrote the manuscript with feedback from all the authors.

## Competing interests

R.P. is currently an employee at BioNTech SE (Mainz, Germany). All the other authors declare no conflicts of interest.
