## [Peer Review File · Nature Communications]

REVIEWER COMMENTS

Reviewer #1 (Remarks to the Author):

The manuscript describes a systematic study of lipid-based nanoparticles that are aimed to be used as drug delivery systems, more specifically, delivery of siRNA. The study involves systems both with and without siRNA and they are studied by cryo-TE, cryo-ET (electron tomography), and SAXS. Investigations of thermal stability, time development and transfection efficiency are also carried out. Various liquid crystalline phases are identified and it is found that systems, which have a persistent hexagonal phase are most efficient with respect to transfection, probably because they completely avoid the lamellar phase that loses siRNA. The conclusions are well justified.

Cryo-TEM and cryo-EM are used for rather detailed structural studies of some of the structures and SAXS is used to supplement these. Time development and thermal stability is only really studied by SAXS as this is an in situ technique, which is easy to use for such investigations. However, the SAXS data are somewhat under-analyzed and under-exploited as only peak positions are determined and only qualitative interpretation is done. Whether the data contains more information is difficult to assess because a linear intensity scale is used. More subtle variations in the low intensity range are much easier to see with a logarithmic scale and therefore the authors are encouraged to change the representation in the plots. More quantitative analysis in terms of domain sizes, disorder parameters and cross section structure may be possible applying the theoretical framework of Forster et al J. Phys. Chem. B 2005, 109, 4, 1347–1360, <https://doi.org/10.1021/jp0467494>

I am puzzled about the use of the term 'paracrystalline'. The usual term to describe the order formation in systems of the kind described in the manuscript is (lyotropic) 'liquid crystalline', where some of the degrees of freedom display liquid-like order. Whereas liquid crystals are fairly well defined in 1, 2 and 3 dimensions, paracrystals are not, although some attempts have been made in recent years, if the authors have a special reason for using the term paracrystal, they should give it in the manuscript or otherwise use the usual term.

Reviewer #2 (Remarks to the Author):

In this study, the authors develop a bottom-up approach to assemble and study lipid nanoparticles (LNPs) that can entrap, protect, and deliver small interfering RNAs (siRNAs). Specifically, the authors assemble canonical DODAP LNPs entrapping siRNA cargoes; cryogenic transmission electron microscopy, cryogenic electron tomography, and small-angle X-ray scattering data are then collected for each formulation with the goal of analyzing LNP structure (with a key focus on identifying lamellar vs. inverse hexagonal structures as a function of LNP formulation, the structure of filled vs. empty LNPs, and the role that temperature plays on influencing LNP structure). Finally, in vitro dose-response curves that evaluate the efficacy of each LNP formulation with one type of siRNA (eGFP) are reported on two different cell lines with the goal of understanding the importance of LNP structure (ex. lamellar vs hexagonal) on efficacy, and model studies evaluating the mechanistic interaction of these siRNA LNPs on large unilamellar vesicles (LUVs) are also performed as a model to verify which LNP structural components are important for uptake in their model uptake system.

Overall, the manuscript is well-written and well-organized, both of which will aid in clear and strong reader comprehension. The imaging analysis of the LNPs is also strong and thorough, and the selected images clearly delineate the points the authors are trying to make in terms of correlating the LNP formulations to their overall structure. The in text descriptions of the data is also strong and clear.

However, this reviewer believes that there are a number of major and minor considerations that should be addressed including additional experiments to strengthen the report. These suggestions are provided below for the authors' consideration:

Major:

1. Figure 5b/5c: Statistical analysis should be performed and represented on each graph to which differences between the 10PE-D5-LNP-NP1, 10PE-LNP-NP1, 30PE-LNP-NP1, and 49PE-LNP-NP1 are statistically significant

2. Figure 5. In the text it is noted that "In order to monitor the difference in uptake between formulations, a small amount of the non-exchangeable lipid dye 1,1'-dioctadecyl-3,3,3'-tetramethylindodicarbocyanine (DiD, 0.1 mol%) was added to the formulations during formation." These formulations are then evaluated for siRNA mediated GFP silencing following administration onto two different cell populations in a dose responsive fashion, ultimately correlating the LNP structures described in the previous imaging data to the silencing efficacy of the particles.

It is the opinion of this reviewer that this study should be improved because the LNP formulations that were imaged are not the same as the LNP formulations that are being evaluated in vitro (due to the lack of the DiD dye in the imaging data, and the presence of the DiD dye in the in vitro studies). Accordingly, the imaging data for the DiD-incorporating LNP formulations should be collected, provided, and compared to ensure that the incorporation of this dye does not impact the structure of the base LNP formulation.

3. Figure 5. Building on revision 1 above, it is the recommendation of this reviewer that the cell culture studies be repeated in full with the non-DiD incorporating LNP formulations. Specifically, this study would analyze the GFP silencing of each non-DiD incorporating LNP formulation (without monitoring the uptake differences). The reason for this recommendation is because dye molecules can influence the uptake and/or efficacy of RNA LNP particles, and this control experiment should therefore be performed to eliminate this as a potential variable.

4. Figure 6. In this experiment, the authors study "the interaction of LNPs with anionic large unilamellar vesicles (LUVs) that mimic endosomal membranes". The goal of this experiment is to "gain more mechanistic insight into the LNP-membrane interaction". This reviewer agrees that understanding the mechanism of these particles is central to the manuscript, particularly towards the authors' goal of correlating LNP structure to activity.

However, it is the opinion of this reviewer that this experiment is not ideal for achieving this goal given that LUVs are considerably simpler than living cells which limits the ability to extrapolate this data set in a meaningful capacity towards cellular uptake. To compliment this data set and add validity to their

models, this reviewer recommends performing confocal imaging data with the studied LNP formulations to determine if the trends observed with the LUVs match those observed in confocal imaging studies.

5. Building on these points, it is the opinion of this reviewer that the authors' work would be considerably strengthened if they demonstrated that their structure-activity relationships hold for more than one type of RNA cargo. For example, it is recommended that the authors could perform an identical suite of imaging data and cell culture efficacy and tolerability studies with mRNAs encoding for GFP (or a different reporter protein).

6. Further building on these points, it is the opinion of this reviewer that the authors' work would be strengthened if they demonstrated that their structure-activity relationships hold for clinically-relevant LNP formulations including those that form the basis of the Biontech – Pfizer and Moderna Covid19 vaccines.

7. Further, this reviewer believes that it would be important to study how incubation of these particles in serum influences the LNP structure using the authors' imaging techniques. Serum proteins that adsorb to the surface of LNPs may influence their uptake, and it is the opinion of this reviewer that analyzing the effect (if any) that serum protein adsorption has on LNP structure would strengthen this report.

8. Given that the LNPs described in this report would have therapeutic application in vivo (for example, the siRNA molecule studied in this report is Patisiran which is used in the clinically approved LNP formulation Onpattro), it is the opinion of this reviewer that this report would benefit from an intravenous dosing study in mice to observe if the same structure-activity relationships observed in their in vitro studies match those that are observed in vivo.

Minor:

1. Page 3, Line 50: The word "cationic" should be removed to be consistent with the nomenclature in the remainder of the manuscript
2. Page 4, Line 72: The order of the words "these" and "how" should be inverted
3. Page 4, Line 77: The word "relationship" should be pluralized to "relationships"
4. Figure 1g,h,i: To ease reader comprehension, the authors may consider defining the white vs. black arrows into the Figure caption legend as is done in the legend for Figure 4

Reviewer #3 (Remarks to the Author):

This paper presents structural characterization of LNP of varying composition using cryoTEM, cryoET and small angle X-ray scattering (SAXS). They identified lamellar, inverted hexagonal or mixed lipid-RNA structures by changing the DOPE lipid content and ionizable lipid to siRNA ratios. They showed that the composition with the highest PE content (49PE-LNP-NP1) had an inverted hexagonal structure and showed the most GFP silencing across all doses.

PE is known to be fusogenic due to its conical effective molecular shape and tendency to form an inverted hexagonal H(II) phase. It is frequently used in cationic lipid formulations for gene delivery. Importantly, LNP formulations including PE has already been produced, and its efficacy in destabilising

endosomal membranes and enabling endosomal escape has been demonstrated. Kuffman et al. (Nano Lett. 15, 7300–7306 (2015), for example, reported on an optimised formulation of LNP. Similar to this work, the key features of their optimization were the incorporation of DOPE and increased ionizable lipid:mRNA weight ratios.

The lipid mixing assay detects lipid transfer from the donor to the acceptor membrane but does not distinguish between fusion, hemifusion, or lipid exchange without membrane sharing. Differentiating between these processes necessitates further experiments, such as content mixing.

Furthermore, the lipid mixing experiment provides no information on the disintegration of LNP or the release of RNA after mixing with endosomal mimicking liposomes. FRET tests should be performed to demonstrate LNP dissociation/RNA release by incorporating FRET pairs into the LNP. It should be highlighted that the efficiency of transfection is determined by LNP dissociation and RNA release rather than lipid mixing. Indeed, Cryo-TEM (Fig 6 f and g) shows that the LNP and liposomes are almost intact (not ruptured) even after 7 hrs.

Endosomal escape continues to be a major impediment to the advancement of RNA-based therapeutics. To overcome this barrier, the precise process of nanocarrier endosomal escape must be understood. Although this paper gives an interesting structural analysis, however the effect of PE is not new and I believe it is only of interest to a very specific audience and does not justify publication in Nature Communications.

Reviewer #4 (Remarks to the Author):

Pattipeiluhu and colleagues report on morphologies of siRNA lipid nanoparticles enhancing transfection efficiency. In a first part of the ms, the authors performed structural characterizations of three formulations using SAXS and CryoEM. They concluded that an increase of PE changes the lipid organization from lamellar to inverted hexagonal phase within the nanoparticles. In a second part, they observed that increasing the temperature from 25°C to 37°C induces a disorder of the lamellar phase. These three formulations are used to perform intracellular silencing of eGFP. The 49PE-LNP-NP1 formulation appears to be the best candidate for inducing gene silencing. To get clue on the role of PE in the endosomal escape pathway, the authors analyzed low pH effect in the presence of anionic liposomes using lipid mixing experiments, cryoEM and SAXS.

General comments

The authors provide evidence that the nature of lipid organization for nucleic acid delivery is important. The role of PE in generating non-lamellar phase improves the efficacy of intracellular gene silencing. Because of the effect of the temperature on the lipid organisation, the characterization of nanoparticles at 37°C is more appropriate than that at 25°C. The characterization of the nanoparticles at 25°C should be less detailed in the ms. The report will be strengthened if the focus was given on the nanoparticle characterization at 37°C and at pH 6.

A lot of effort has been spent on analysing the morphology of 49PE-LNP-NP1 nanoparticles and the regular spacing. The authors claim that lipid adopts an inverted hexagonal phase that favours both lipid mixing and gene silencing. However, line 340, the authors indicate that these LNP are polymorphic. It is

rather confusing because it is not clear whether the enhancement of lipid mixing and gene silencing is due to the inverted hexagonal phase or not. More generally, it is known that lipid can adopt various phases specially depending on temperature. In addition to the hexagonal phase, cubic phase could be generated. The authors should comment on that.

The temperature drastically changes the lipid organisation for 10PE-LNP-NP1 as shown in figure 4I. It is partly lamellar and also amorphous. However, in fig 6F, the 10PE-LNP-NP1 organisation looks as hexagonal phase. They both were submitted to a 37°C treatment. Why do they look so different ?

Lipid mixing is enhanced at pH6 even for 49PE-LNP-NP1 . Is there a change in lipid organization that could explain this effect?

CryoEM images are convincing, but tomographic data are not so well described. It is hard for the reader to figure out whether these data are relevant or not (see below my comments).

Additional comments

-Figure 2a it seems that the small features between lipid bilayers could be siRNA. It is possible to compare with LNP-noRNA. Is there a way to see siRNA densities?

-Figure 2C. 10PE-LNP-NP1 particles are composed of concentric lipid layers. The proposed model does not reflect this concentric organization. It must be clarified.

-Line 216 the argument to explain the deviation of lattice spacing is not clear.

- 10PE-LNP-noRNA particles appear different from 10PE-LNP-NP1 particles. They are less concentric and even exhibit some hexagonal shape. How authors explain this difference ? what is the buffer for particles preparation without siRNA?

- Figure 3D shows tomogram sections. Authors should explain how to interpret the densities. what is the slice thickness? What are the spherical and tubular structures (line 368)? They correspond to which lipid phase?. This should be clarified.

-Figure 3E shows a reconstruction. It looks like a drawing model and not a real 3D reconstruction. Authors must give more details.

-Figure 3F shows tomogram sections of nanoparticle with siRNA. As for fig3D , how to interpret the densities? Do we see siRNA? Further comments are needed. Is the spacing identical to the one measured on the 2D image? Authors should comment also on the effect of missing wedge whether or not there are some structural bias.

- Figure 3G seems rather poor in term of relevant information compared with Figure3F. And additional effort should be done to help the reader. Fig3H show interesting features but details are missing how it has been produced. Authors should explain the fitting method.

Figure 5: A control with scrambled RNA is missing to validate the effect of siRNA.

Figure 6F,G, The pH value is missing Figure 6H The model for the mechanism is incorrect. At 37°C, LNP with lamellar structure does not exit according to the results. Likewise for the scheme in fig6A . Few sentences are confusing in the legend of figure 6

REVIEWER COMMENTS

Reviewer #1 (Remarks to the Author):

The manuscript describes a systematic study of lipid-based nanoparticles that are aimed to be used as drug delivery systems, more specifically, delivery of siRNA. The study involves systems both with and without siRNA and they are studied by cryo-TE, cryo-ET (electron tomography), and SAXS. Investigations of thermal stability, time development and transfection efficiency are also carried out. Various liquid crystalline phases are identified and it is found that systems, which have a persistent hexagonal phase are most efficient with respect to transfection, probably because they completely avoid the lamellar phase that loses siRNA. The conclusions are well justified. Cryo-TEM and cryo-EM are used for rather detailed structural studies of some of the structures and SAXS is used to supplement these. Time development and thermal stability is only really studied by SAXS as this is an in situ technique, which is easy to use for such investigations. However, the SAXS data are somewhat under-analyzed and under-exploited as only peak positions are determined and only qualitative interpretation is done. Whether the data contains more information is difficult to assess because a linear intensity scale is used. More subtle variations in the low intensity range are much easier to see with a logarithmic scale and therefore the authors are encouraged to change the representation in the plots.

We thank the reviewer for claiming that the conclusions given in the manuscript are well justified. Regarding the use of logarithmic scales, we agree that they are more commonly used for displaying SAXS data in the literature. Therefore, we have changed the axes of the SAXS data throughout the manuscript from a linear-linear to logarithmic-logarithmic. Nevertheless, since the only data used from the SAXS curves are the positions corresponding to the Bragg peaks, the change of this representation does not impact the values, the analysis and the conclusions derived thereof.

More quantitative analysis in terms of domain sizes, disorder parameters and cross section structure may be possible applying the theoretical framework of Forster et al J. Phys. Chem. B 2005, 109, 4, 1347–1360, <https://doi.org/10.1021/jp0467494>

We agree with the reviewer that SAXS allows for more in-depth (theoretical) analysis, for example looking at sample morphology and size. Nevertheless, these types of analyses are in our case troubled by the co-existence of multiple phases within most of the samples that we have studied. For example, lower limit diffraction values describing information about particle morphology or size are problematic for the polyphasic particles. In this sense we believe that the analysis of identified Bragg peaks and their correlation to observations obtained from cryoEM analysis are considered as most relevant, and can still be accurately applied despite the occurrence of co-existence. Therefore, further analysis of SAXS curves is omitted in this manuscript.

I am puzzled about the use of the term 'paracrystalline'. The usual term to describe the order formation in systems of the kind described in the manuscript is (lyotropic) 'liquid crystalline', where some of the degrees of freedom display liquid-like order. Whereas liquid crystals are fairly well defined in 1, 2 and 3 dimensions, paracrystals are not, although some attempts have been made in recent years. If the authors have a special reason for using the term paracrystal, they should give it in the manuscript or otherwise use the usual term.

We agree with the reviewer's comment regarding the term liquid crystalline and have replaced all the terminology from "paracrystalline" to "liquid crystalline" in the title and throughout the manuscript and supplementary information.

Reviewer #2 (Remarks to the Author):

In this study, the authors develop a bottom-up approach to assemble and study lipid nanoparticles (LNPs) that can entrap, protect, and deliver small interfering RNAs (siRNAs). Specifically, the authors assemble canonical DODAP LNPs entrapping siRNA cargoes; cryogenic transmission electron microscopy, cryogenic electron tomography, and small-angle X-ray scattering data are then collected for each formulation with the goal of analyzing LNP structure (with a key focus on identifying lamellar vs. inverse hexagonal structures as a function of LNP formulation, the structure of filled vs. empty LNPs, and the role that temperature plays on influencing LNP structure). Finally, in vitro dose-response curves that evaluate the efficacy of each LNP formulation with one type of siRNA (eGFP) are reported on two different cell lines with the goal of understanding the importance of LNP structure (ex. lamellar vs hexagonal) on efficacy, and model studies evaluating the mechanistic interaction of these siRNA LNPs on large unilamellar vesicles (LUVs) are also performed as a model to verify which LNP structural components are important for uptake in their model uptake system.

Overall, the manuscript is well-written and well-organized, both of which will aid in clear and strong reader comprehension. The imaging analysis of the LNPs is also strong and thorough, and the selected images clearly delineate the points the authors are trying to make in terms of correlating the LNP formulations to their overall structure. The in-text descriptions of the data are also strong and clear.

We kindly thank the reviewer for thorough evaluation of the manuscript and the positive and constructive feedback.

However, this reviewer believes that there are a number of major and minor considerations that should be addressed including additional experiments to strengthen the report. These suggestions are provided below for the authors' consideration:

Major:

1. Figure 5b/5c: Statistical analysis should be performed and represented on each graph to which differences between the 10PE-D5-LNP-NP1, 10PE-LNP-NP1, 30PE-LNP-NP1, and 49PE-LNP-NP1 are statistically significant

We agree with the reviewer that statistical analysis of the shown curves would strengthen the conclusion on having significantly different behaviors between LNP formulations. However, although replicate measurements for each concentration are performed, simple statistical difference between points on a dose-response curve is typically not performed as it does not give relevant information for the experiment. In this case, we can perform two-way ANOVA in which the statistical significance of the entire curves can be determined, yielding a p value < 0.0001. The authors feel that what might generate more relevant statistical information is the determination of the relevant IC₅₀-values with a 95% confidence interval derived from the dose-response curves to more intuitively display that there is a clear increase in potency for formulations with increasing %DOPE. The calculated values and their deviations are added to the supplementary information as Supplementary Table 3 as well as relevantly throughout the main text and in the figure captions.

2. Figure 5. In the text it is noted that “In order to monitor the difference in uptake between formulations, a small amount of the non-exchangeable lipid dye 1,1'-dioctadecyl-3,3,3'-tetramethylindodicarbocyanine (DiD, 0.1 mol%) was added to the formulations during formation.” These formulations are then evaluated for siRNA mediated GFP silencing following administration onto two different cell populations in a dose responsive fashion, ultimately correlating the LNP structures described in the previous imaging data to the silencing efficacy of the particles. It is the opinion of this reviewer that this study should be improved because the LNP formulations that were imaged are not the same as the LNP formulations that are being evaluated in vitro (due to the lack of the DiD dye in the imaging data, and the presence of the DiD dye in the in vitro studies). Accordingly, the imaging data for the DiD-incorporating LNP formulations should be collected, provided, and compared to ensure that the incorporation of this dye does not impact the structure of the base LNP formulation.

The response to this point has been given in conjunction with that to point 3 (see below), as they address a similar topic.

3. Figure 5. Building on revision 1 above, it is the recommendation of this reviewer that the cell culture studies be repeated in full with the non-DiD incorporating LNP formulations. Specifically, this study would analyze the GFP silencing of each non-DiD incorporating LNP formulation (without monitoring the uptake differences). The reason for this recommendation is because dye molecules can influence the uptake and/or efficacy of RNA LNP particles, and this control experiment should therefore be performed to eliminate this as a potential variable.

Response to point 2 and 3: The authors do agree with the reviewer's comment that additional controls should be performed to determine if there is no significant change in structures when incorporating these molecules. To this end, we have performed cryoTEM imaging of 49PE-LNP-NP-NP1 containing 0.1 mol% of DiD fluorescent dye, showing similar hexagonal structure and liquid crystalline spacing (~6.3 nm) as their control displayed in the main text of the manuscript. This has been added as Supplementary Figure 13. In addition, we have performed a control cellular transfection experiment in the HeLa-CD63-eGFP cell line. Here, we compared 49-PE-LNP-NP1 without DiD and including DiD. Here, it is shown that the dose-response of both formulations is very similar, indicating no influence of the low fraction of DiD on the cellular transfection. In addition, we tested 49PE-LNP-NP1 encapsulating a negative control scrambled siRNA for eGFP. The absence of any transfection across the entire dosing titration demonstrates that there is no LNP-induced or dye-induced background transfection occurring. This data has been added as Supplementary Figure 15. The sequence of negative control siRNA was obtained commercially, and the sequence has been added to Supplementary Table 1.

4. Figure 6. In this experiment, the authors study “the interaction of LNPs with anionic large unilamellar vesicles (LUVs) that mimic endosomal membranes”. The goal of this experiment is to “gain more mechanistic insight into the LNP-membrane interaction”. This reviewer agrees that understanding the mechanism of these particles is central to the manuscript, particularly towards the authors' goal of correlating LNP structure to activity.

However, it is the opinion of this reviewer that this experiment is not ideal for achieving this goal given that LUVs are considerably simpler than living cells which limits the ability to extrapolate this data set in a meaningful capacity towards cellular uptake. To compliment this data set and add validity to their models, this reviewer recommends performing confocal

imaging data with the studied LNP formulations to determine if the trends observed with the LUVs match those observed in confocal imaging studies.

The motivation behind the choice of LUVs with a size of 150 nm is to mimic the interactions with LNPs and endosomal membranes. In the case of ionizable LNPs, the main mechanistic feature is that they interact with these membranes after endocytosis due to an electrostatic interaction that is triggered upon acidification of the endosomes. To this end, the scale of 100-300 nm and the pH of 6.0 is considered as most relevant.¹

In response to potential confocal imaging studies, the LUVs (~150 nm) are below the diffraction limit of light and therefore confocal imaging cannot be applied to resolve these or any interactions between LNPs and the LUVs. In addition, using light-microscopy techniques, the structural details that are derived from the cryoEM studies of LNPs with model LUVs cannot be obtained.

5. Building on these points, it is the opinion of this reviewer that the authors' work would be considerably strengthened if they demonstrated that their structure-activity relationships hold for more than one type of RNA cargo. For example, it is recommended that the authors could perform an identical suite of imaging data and cell culture efficacy and tolerability studies with mRNAs encoding for GFP (or a different reporter protein).

We chose siRNA as the structures are unchanged between sequences, and so our data can be more broadly applied, as we state on lines 100-102. In contrast, mRNA molecules show a much broader difference between them in terms of sequence, lengths as well as secondary and tertiary structures. Therefore, we believe that a systematic study is not possible for all different RNA sequences and lengths would be outside of the scope of this manuscript.

6. Further building on these points, it is the opinion of this reviewer that the authors' work would be strengthened if they demonstrated that their structure-activity relationships hold for clinically-relevant LNP formulations including those that form the basis of the Biontech – Pfizer and Moderna Covid19 vaccines.

7. Further, this reviewer believes that it would be important to study how incubation of these particles in serum influences the LNP structure using the authors' imaging techniques. Serum proteins that adsorb to the surface of LNPs may influence their uptake, and it is the opinion of this reviewer that analyzing the effect (if any) that serum protein adsorption has on LNP structure would strengthen this report.

8. Given that the LNPs described in this report would have therapeutic application in vivo (for example, the siRNA molecule studied in this report is Patisiran which is used in the clinically approved LNP formulation Onpattro), it is the opinion of this reviewer that this report would benefit from an intravenous dosing study in mice to observe if the same structure-activity relationships observed in their in vitro studies match those that are observed in vivo.

Response to points 5-8: The authors agree with the reviewer that the raised points will be interesting and worthwhile topic for further exploration of the system, but that they fall outside of the scope of this manuscript. Regarding point 7, we would like to add that the cell culture experiments were done in the presence of serum proteins, and that there is a clear correlation between cell culture silencing, observed structures and lipid mixing in the FRET assay. Although the absorption of serum proteins is likely to happen, a more extensive analysis involving mass spectrometry techniques would have to be utilized to get in-depth and useful information on the exact mechanisms here.

Minor:

1. Page 3, Line 50: The word “cationic” should be removed to be consistent with the nomenclature in the remainder of the manuscript
2. Page 4, Line 72: The order of the words “these” and “how” should be inverted
3. Page 4, Line 77: The word “relationship” should be pluralized to “relationships”
4. Figure 1 g,h,i: To ease reader comprehension, the authors may consider defining the white vs. black arrows into the Figure caption legend as is done in the legend for Figure 4

The authors agree with the reviewer and have processed all the minor comments as described above

Reviewer #3 (Remarks to the Author):

This paper presents structural characterization of LNP of varying composition using cryoTEM, cryoET and small angle X-ray scattering (SAXS). They identified lamellar, inverted hexagonal or mixed lipid-RNA structures by changing the DOPE lipid content and ionizable lipid to siRNA ratios. They showed that the composition with the highest PE content (49PE-LNP-NP1) had an inverted hexagonal structure and showed the most GFP silencing across all doses. PE is known to be fusogenic due to its conical effective molecular shape and tendency to form an inverted hexagonal H(II) phase. It is frequently used in cationic lipid formulations for gene delivery. Importantly, LNP formulations including PE has already been produced, and its efficacy in destabilising endosomal membranes and enabling endosomal escape has been demonstrated. Kuffman et al. (Nano Lett. 15, 7300–7306 (2015), for example, reported on an optimised formulation of LNP. Similar to this work, the key features of their optimization were the incorporation of DOPE and increased ionizable lipid:mRNA weight ratios.

The authors kindly thank the reviewer for the evaluation of the manuscript and providing with positive and constructive feedback. In response to the cited article from Kuffman et al., we acknowledge that the paper does indeed mention the optimization of DOPE usage for mRNA delivery but does not find similar behavior for siRNA delivery. In this case, it is hard to determine what might be the potential root-cause, as many different factors differ here such as chosen lipids and compositions. We find that our work, in which thorough structural analysis is coupled to the transfection behavior, gives a clear and rational root cause. The information generated here, could be used in further studies such as those from Kuffman et al. to couple the structural identity of their successful formulations to biological behavior.

The lipid mixing assay detects lipid transfer from the donor to the acceptor membrane but does not distinguish between fusion, hemifusion, or lipid exchange without membrane sharing. Differentiating between these processes necessitates further experiments, such as content mixing. Furthermore, the lipid mixing experiment provides no information on the disintegration of LNP or the release of RNA after mixing with endosomal mimicking liposomes. FRET tests should be performed to demonstrate LNP dissociation/RNA release by incorporating FRET pairs into the LNP. It should be highlighted that the efficiency of transfection is determined by LNP dissociation and RNA release rather than lipid mixing. Indeed, Cryo-TEM (Fig 6 f and g) shows that the LNP and liposomes are almost intact (not ruptured) even after 7 hrs.

The authors agree that the lipid mixing assay does have certain limitations in terms of giving detailed information on RNA release across a membrane. However, this is why lipid mixing was used a model to study the initial interaction of the LNPs with an anionic membrane and

to determine if an immediate switch in structural phases is observed (in a timeframe of hours). To then determine the effectiveness of RNA release, we have resorted to cellular transfection. The limitation of optimizing further model lipid mixing and fusion systems is that it would require the introduction of multiple fluorescent probes into the nanoparticles which might alter their behavior significantly. In addition, the dynamics after the initial interaction cannot be properly modeled due to the lack of biologically relevant molecules such as enzymes and membrane dynamics which are believed to play a critical role in nanoparticle transfection. Therefore, we believe that the use of our model, aided by multiple lines of evidence coming from SAXS and cryo-EM analysis, provides a unique in-sight of LNP-membrane interaction in the practically relevant timeframe.

In regard to the point that addresses that LNPs are mostly intact upon interaction with the liposomes, this behavior is well reflected by an in-depth approach of similar nanoparticles in which it was found that only a small percentage of the RNA molecules (1-3%) are able to enter into the cytosol. We believe that this unique in-sight on LNP-membrane interaction has not been visualized in such an extent before and can help further studies to design more efficient of lipid-based nanocarriers in such a way that they fully fuse with acceptor membranes.

Endosomal escape continues to be a major impediment to the advancement of RNA-based therapeutics. To overcome this barrier, the precise process of nanocarrier endosomal escape must be understood. Although this paper gives an interesting structural analysis, however the effect of PE is not new and I believe it is only of interest to a very specific audience and does not justify publication in Nature Communications.

This is the first example in which we can clearly determine the internal structure of a siRNA-LNP and that by tuning lipid composition the internal structure changes, which affects the delivery efficiency in cells. This manuscript is a conceptual biophysical study rather than finding the best ever LNP formulation. To date all approved LNP formulations are obtained by brute force/massive screening, while our manuscript is an attempt to bring rationale into the field of LNP design.

Reviewer #4 (Remarks to the Author):

Pattipeiluhu and colleagues report on morphologies of siRNA lipid nanoparticles enhancing transfection efficiency. In a first part of the ms, the authors performed structural characterizations of three formulations using SAXS and CryoEM. They concluded that an increase of PE changes the lipid organization from lamellar to inverted hexagonal phase within the nanoparticles. In a second part, they observed that increasing the temperature from 25°C to 37°C induces a disorder of the lamellar phase. These three formulations are used to perform intracellular silencing of eGFP. The 49PE-LNP-NP1 formulation appears to be the best candidate for inducing gene silencing. To get clue on the role of PE in the endosomal escape pathway, the authors analyzed low pH effect in the presence of anionic liposomes using lipid mixing experiments, cryoEM and SAXS.

We kindly thank the reviewer for thorough evaluation of the manuscript and the positive and constructive feedback.

General comments

The authors provide evidence that the nature of lipid organization for nucleic acid delivery is

important. The role of PE in generating non-lamellar phase improves the efficacy of intracellular gene silencing. Because of the effect of the temperature on the lipid organisation, the characterization of nanoparticles at 37°C is more appropriate than that at 25°C. The characterization of the nanoparticles at 25°C should be less detailed in the ms. The report will be strengthened if the focus was given on the nanoparticle characterization at 37°C and at pH 6.

The temperature and pH at which nanoparticle characterization is performed classically throughout literature is at room temperature (25 °C) and the relevant pH for the final storage matrix (in this case pH 7.4). This is the case, as LNPs are evaluated for extended characterization after manufacturing and fill-and-finish,² which in this text will be referred to as the t0-point. We believe that the examination of biologically relevant parameters (37 °C, pH < 7) in our manuscript is done in a rational and sufficient manner. As these parameters change only upon the administration of the nanoparticles to the biological system, we have investigated the dynamic change from a t0-point to one that would occur *in situ* exclusively. As for initial temperature change upon administration to the system, we have stated on page 21, line 400: "By contrast, neither 30PE-LNP-NP1 nor 49PE-LNP-NP1 exhibited a significant change in their SAXS profiles collected at either 25 °C or 37 °C". For the 10PE-LNP-NP1, which does change upon heating, we have provided extensive characterization with SAXS and cryo-EM. In addition, the change of both temperature 37 °C and pH (=6) have been addressed in the lipid mixing model. All together, we believe that a strong t0-point analysis is necessary to compare and evaluate our LNPs after manufacturing and to literature, while the dynamic changes due to changes in temperature, pH and exposure to other lipid membranes give us a unique in-depth understanding of their behavior.

A lot of effort has been spent on analysing the morphology of 49PE-LNP-NP1 nanoparticles and the regular spacing. The authors claim that lipid adopts an inverted hexagonal phase that favours both lipid mixing and gene silencing. However, line 340, the authors indicate that these LNP are polymorphic. It is rather confusing because it is not clear whether the enhancement of lipid mixing and gene silencing is due to the inverted hexagonal phase or not. More generally, it is known that lipid can adopt various phases specially depending on temperature. In addition to the hexagonal phase, cubic phase could be generated. The authors should comment on that.

We agree that the use of polymorphic in the sense of 49PE-LNP-NP1 might be considered as confusing. Here, polymorphic was intended to be used to describe the division between ordered structures and the rest of the material that fills up the remaining of the particles. Here, we were not able to obtain clear structural ordering with cryoEM and the presence of possible invariant structures does not explain the differences in mixing and delivery, we consequently focus on the dominant, varying and controllable changes in morphology of the inverse hexagonal phases. In the case of the 49-LNP-noRNA formulation, it describes the difference between the observed hexagonal structures and inverse micelles phases (Supplementary Figures 10 and 11).

We agree that there might be a potential formation of the cubic phase, as it has been previously linked to the DOPE containing lipid-based nanosystems.^{3,4} In this case, our multiple lines of evidence coming from cryo-EM and SAXS for all systems help us deduce that those phases are not observed.

- The observed fast Fourier transform analyses for all observed structures either reflect hexagonal spacings or straight lines, and none similar to those expected for cubic phases for example described in ref

- The Braggs peaks observed in the SAXS profiles can only be directly linked to the cryo-EM spacing values if a hexagonal phase model is, and not when the cubic model is applied (see table below and Supplementary Figure 10). For example, in the case of tubular structures the cryoEM derived values were 4.90 nm (49PE-LNP-noRNA) and 6.30 nm (49PE-LNP-NP1) on average. Here, the d-spacing from the hexagonal model reflects this highly accurately with values of 4.95 nm and 6.22 nm respectively, whereas the cubic model leads to indifferent values of 6.99 nm and 8.80 nm respectively.
- Application of the cubic phase also does not lead to identical values obtained for the first order and second-order peak. In the case of the inverse hexagonal, the first and second order values are very similar (8.39 nm and 8.57 nm in the case of 49PE-LNP-noRNA and 10.65 nm and 10.78 nm for 49PE-LNP-NP1).

Altogether, this helps us conclude that inverse hexagonal structures accurately describe the observed structures. Nevertheless, it might be that cubic phases could be present in other formulations containing DOPE and they should in the future be evaluated properly.

$$d_{hex} = \frac{2\pi n}{q}, \text{ with } n = n[1,0] = 1, n[1,1] = \sqrt{3}$$

$$d_{cub} = \frac{2\pi n}{q}, \text{ with } n = n[1,0] = \sqrt{2}, n[1,1] = \sqrt{3}$$

Sample	Structure	CryoEM value average (nm)	q_{max} value (Å)	Braggs peak	d-spacing SAXS hexagonal model (nm)	d-spacing SAXS cubic model (nm)
49PE-LNP-noRNA	Inverse hexagonal spheres	7.50 – 8.15	0.0749	n[1,0]	8.39	11.86
			0.127	n[1,1]	8.57	8.57
	Inverse hexagonal tubes	4.90	0.127	n[1,0]	4.95	6.99
49PE-LNP-NP1	Polymorphic	Not determined	0.0590	n[1,0]	10.65	15.0
			0.101	n[1,1]	10.78	10.78
	Inverse hexagonal tubes	6.30	0.101	n[1,0]	6.22	8.80

The above table has been added to the Supplementary information as supplementary Table 4. In addition, the text in the main text has been edited to address this analysis and is described in lines 346-352.

The temperature drastically changes the lipid organisation for 10PE-LNP-NP1 as shown in figure 4I. It is partly lamellar and also amorphous. However, in fig 6F, the 10PE-LNP-NP1 organisation looks as hexagonal phase. They both were submitted to a 37°C treatment. Why do they look so different?

In Figure 6 it is not the pure LNPs that have been imaged. In this case the LNPs have been mixed with LUVs at a pH of 6.0, leading to an interaction between the LNPs and LUVs. What can be seen are combined structures of LUVs (aqueous core similar to Supplementary Figure 16) and the electron dense structures of the LNPs.

Lipid mixing is enhanced at pH6 even for 49PE-LNP-NP1 . Is there a change in lipid organization that could explain this effect?

CryoEM images are convincing, but tomographic data are not so well described. It is hard for the reader to figure out whether these data are relevant or not (see below my comments).

We thank the reviewer for suggesting these additional descriptions. We have addressed their points below.

Additional comments

-Figure 2a it seems that the small features between lipid bilayers could be siRNA. It is possible to compare with LNP-noRNA. Is there a way to see siRNA densities?

-Figure 2C. 10PE-LNP-NP1 particles are composed of concentric lipid layers. The proposed model does not reflect this concentric organization. It must be clarified.

-Line 216 the argument to explain the deviation of lattice spacing is not clear.

We have simplified this description as we did not feel it added to the main narrative of the paper. Consequently, we have removed the proposed model and discussion of the deviation, which we have now attributed to a range of structures with one that predominates, which correlates with the SAXS data.

New text on page 11: *“Analyzing all of the selected single particles allowed us to construct a violin plot, revealing the variation in lamellar lattice spacing which range from 4.20 nm to 6.90 nm (Figure 2b). This range is not well reflected in the SAXS profile of 10PE-LNP-NP1, which displays a narrow peak at $q \sim 0.1 \text{ \AA}^{-1}$ (Figure 1c). However, the median lattice spacing of 6.18 nm predominates, presumably causing the narrow peak observed by SAXS. Furthermore, this value closely reflects the expected values of lamellar lipid bilayers containing (phospho)lipids with similar carbon tail lengths (C18) and cholesterol complexing oligonucleotides.”*

- 10PE-LNP-noRNA particles appear different from 10PE-LNP-NP1 particles. They are less concentric and even exhibit some hexagonal shape. How authors explain this difference? what is the buffer for particles preparation without siRNA?

The buffer is identical for all preparations throughout the manuscript. The change of surface morphology of the particles can be explained by the change of ionizable lipid to mRNA ratio. In the case of LNPs without RNA, there is no interaction of charged ionizable lipids with these molecules, allowing all molecules to form LNPs with their desired energy minimum. When siRNA molecules are introduced, especially in the case of NP1 where there is a large stoichiometric amount added, there is an electrostatic interaction occurring between charged ionizable lipids and negative charged siRNA molecules. This in turn drives the formation of lipid-RNA structures which in this case will favor the formation of concentric circle.

- Figure 3D shows tomogram sections. Authors should explain how to interpret the densities. what is the slice thickness? What are the spherical and tubular structures (line 368)? They correspond to which lipid phase?. This should be clarified.

We have now stated the thickness of the slices where appropriate (all were 10 nm thick). The spherical and tubular structures represent the inverse-hexagonally packed empty lipid tubes and micelles, respectively, which we have stated more clearly on line 334.

-Figure 3E shows a reconstruction. It looks like a drawing model and not a real 3D reconstruction. Authors must give more details.

We have added the text: “(e) Reconstruction of the three-dimensional tomogram shown as the middle LNP in d, with manually annotated lipid tubes (yellow) and micellar structures (yellow) within the liposomes volume (pink).” The relevant methods section has been updated with an explanatory reference.

-Figure 3F shows tomogram sections of nanoparticle with siRNA. As for fig3D, how to interpret the densities? Do we see siRNA? Further comments are needed. Is the spacing identical to the one measured on the 2D image? Authors should comment also on the effect of missing wedge whether or not there are some structural bias.

- Figure 3G seems rather poor in term of relevant information compared with Figure3F. And additional effort should be done to help the reader. Fig3H show interesting features but details are missing how it has been produced. Authors should explain the fitting method.

We have added additional text in lines 366-380 and extended the figure legend to more clearly explain these figures. We have also colored the map to aid interpretation.

Fig legend: “(d) CryoET slices of 3 separate 49PE-LNP-noRNA particles. Identified spherical and tubular structures in each slice are indicated as blue circles and yellow lines respectively. Scale bars are 50 nm. (e) Reconstruction of the three-dimensional tomogram shown as the middle LNP in d, with manually annotated lipid tubes (yellow) and micellar structures (yellow) within the liposomes volume (pink). (f) CryoET slices through an individual 49PE-LNP-NP1 particle. EM images represent slices at two different heights of the tomogram, showing straight lines (top) and hexagonally packed regions (bottom). Scale bars are 50 nm. (g) Surface rendering indicates internal liquid crystalline structures (blue) throughout the body of the LNP. The outer membrane is colored pink, as in panel e. (h) Fitted siRNA-lipid models to the electron density derived from cryoET, showing different orientations of the same structure (RNA in purple, lipid tails in yellow).”

Figure 5: A control with scrambled RNA is missing to validate the effect of siRNA.

Experiments with control RNA have been performed and in Supplementary Figure 15, the exact changes are addressed with the comments from other reviewers.

Figure 6F,G, The pH value is missing

The pH value has been added.

Figure 6H The model for the mechanism is incorrect. At 37°C, LNP with lamellar structure does not exit according to the results. Likewise for the scheme in fig6A. Few sentences are confusing in the legend of figure 6

The model for the mechanism in Figure 6H discriminates between a model for the introduction of pre-defined lamellar and pre-defined hexagonally structured LNPs prior to addition of LUVs. As can be seen in Figure 6F, in this scenario, there is still a visible lamellar structure for the 10PE-LNP-NP1 even after 7 hours at 37 °C. We agree with the author that this might seem confusing and have adapted the text accordingly to specify that complete loss of the lamellar structure in these LNPs does not occur when they are mixed with LUVs. This observation highlights another key insight that the separate evaluation of phase transition upon

temperature is not sufficient and that additional evaluation in the presence of LUVs can highlight a more relevant phase transition for biological systems.

References

- (1) Murk, J. L. A. N.; Humbel, B. M.; Ziese, U.; Griffith, J. M.; Posthuma, G.; Slot, J. W.; Koster, A. J.; Verkleij, A. J.; Geuze, H. J.; Kleijmeer, M. J. Endosomal Compartmentalization in Three Dimensions: Implications for Membrane Fusion. *Proc. Natl. Acad. Sci.* **2003**, *100* (23), 13332 LP – 13337.
- (2) Henderson, M. I.; Eygeris, Y.; Jozic, A.; Herrera, M.; Sahay, G. Leveraging Biological Buffers for Efficient Messenger RNA Delivery via Lipid Nanoparticles. *Mol. Pharm.* **2022**, *19* (11), 4275–4285.
- (3) Demurtas, D.; Guichard, P.; Martiel, I.; Mezzenga, R.; Hébert, C.; Sagalowicz, L. Direct Visualization of Dispersed Lipid Bicontinuous Cubic Phases by Cryo-Electron Tomography. *Nat. Commun.* **2015**, *6*, 1–8.
- (4) Leung, S. S. W.; Leal, C. The Stabilization of Primitive Bicontinuous Cubic Phases with Tunable Swelling over a Wide Composition Range. *Soft Matter* **2019**, *15* (6), 1269–1277.

REVIEWER COMMENTS

Reviewer #1 (Remarks to the Author):

I think that the authors have considered all the remarks by the reviewers and made appropriate revisions accordingly. In my opinion, the manuscript can be accepted for publication.

Reviewer #2 (Remarks to the Author):

Thank you. The authors have addressed my points.

Reviewer #3 (Remarks to the Author):

The authors correctly acknowledged the limitation of the lipid mixing assay, which does not provide information about siRNA release. In their response the authors stated that “lipid mixing was used a model to study the initial interaction of the LNPs with an anionic membrane”.

However, there are questions that remain unanswered:

Mechanism underlying lipid-mixing:

The first question is about the mechanism underlying lipid-mixing in the initial interaction phase. Specifically, is it spontaneous lipid transfer, membrane fusion, or hemifusion? Understanding this mechanism is essential to explain why lipid mixing does not proceed to 100% and reaches an equilibrium after approximately 30 minutes for all compositions. Is this observation expected for LNPs?

This is puzzling considering the representative cryoTEM images of 10PE-LNP-NP1 or 49PE-603 LNP-NP1, which show a clear contact between LNP and LUV membranes. One possibility is that lipid mixing occurs only between the outer membrane leaflets of liposomes and LNPs. If this is the case, the question arises whether we can expect siRNA release upon lipid mixing between outer membrane leaflets.

FRET experiment for siRNA release:

It is the opinion of this reviewer that a FRET experiment should be performed to demonstrate siRNA release upon interaction with LUVs. This is important because the proposed mechanisms depicted in Figure 6h suggest siRNA release after lipid mixing. Performing a FRET experiment would provide direct evidence of siRNA release upon interaction with LUVs and support the proposed mechanism. FRET has been employed to monitor the disassembly of LNPs and the subsequent release of siRNA (see, for instance, *AJPS*, 18 (2023) 100769).

Reviewer #4 (Remarks to the Author):

The revised version responds to my concerns.

Reviewer #3 comments/questions:

The authors correctly acknowledged the limitation of the lipid mixing assay, which does not provide information about siRNA release. In their response the authors stated that “lipid mixing was used a model to study the initial interaction of the LNPs with an anionic membrane”. However, there are questions that remain unanswered:

Mechanism underlying lipid-mixing:

The first question is about the mechanism underlying lipid-mixing in the initial interaction phase. Specifically, is it spontaneous lipid transfer, membrane fusion, or hemifusion? Understanding this mechanism is essential to explain why lipid mixing does not proceed to 100% and reaches an equilibrium after approximately 30 minutes for all compositions. Is this observation expected for LNPs? This is puzzling considering the representative cryoTEM images of 10PE-LNP-NP1 or 49PE-603 LNP-NP1, which show a clear contact between LNP and LUV membranes. One possibility is that lipid mixing occurs only between the outer membrane leaflets of liposomes and LNPs. If this is the case, the question arises whether we can expect siRNA release upon lipid mixing between outer membrane leaflets.

FRET experiment for siRNA release:

It is the opinion of this reviewer that a FRET experiment should be performed to demonstrate siRNA release upon interaction with LUVs. This is important because the proposed mechanisms depicted in Figure 6h suggest siRNA release after lipid mixing. Performing a FRET experiment would provide direct evidence of siRNA release upon interaction with LUVs and support the proposed mechanism. FRET has been employed to monitor the disassembly of LNPs and the subsequent release of siRNA (see, for instance, *AJPS*, 18 (2023) 100769).

Response to reviewer #3 comments/questions:

We would like to thank the reviewer for swiftly responding to our revised manuscript, and we are pleased to note that we have been able to answer most of their questions. We are also happy to comment more on the remaining questions and suggestions from the reviewer in which we address 1) The suggested FRET experiment for siRNA release, and 2) the understanding of the mechanism underlying lipid mixing.

1. Suggested FRET experiment for siRNA release

As the reviewer rightfully quoted from our text, we have within this experiment exclusively stated on the possibility of lipid mixing and not on fusion behavior or siRNA release. We have acknowledged that the lipid mixing assay is not able to assess the release of siRNA molecules. To this end, we used the cellular silencing of reporter proteins using siRNA molecules as the most representative and relevant (*i.e.* biological) response for silencing efficiency of the formulations. In the paper that has been cited by the reviewer, the used approached is also an intracellular one,

and does not apply to any model system. Here, fluorescence evaluation of the dye attached to the siRNA is achieved after intracellular processing by the Dicer enzyme. Also, the initial FRET-based fluorescence quenching which is crucial for the experiment is achieved by using gold nanoparticles, something which is not possible within our design. Unfortunately, we are also not aware of any other reliable and technically feasible experiments for studying the siRNA release from LNPs into LUVs. This is most likely because in a similar model system, the release of siRNA molecules from the LNP after interaction would mean that they end up *inside* of the acceptor LUVs, lacking accessibility. In addition, there are a plethora of other conditions at play such as that the nature of the mixing is not aqueous/aqueous and that ionizable lipids will be positively charged and might still facilitate lipid/RNA interactions even after interaction with the LUV. With these points combined, we feel that the current framework in which we have experimentally investigated is the most reliable and insightful as we can get, and that the statements in the text keep these limitations in mind.

2. The understanding of the mechanism underlying lipid mixing

To clarify the mechanism underlying lipid mixing, we would like to give some more explanations. LNPs are solid core lipid nanoparticles which contain a large amount of lipid molecules, especially when compared to a liposome that consists of much fewer lipid molecules and an aqueous core. The classical description of fusion of bilayer membranes, in which a bilayer membrane fuses with another bilayer membrane, is therefore not applicable. Here, as shown in Figure 6b, we show that pH is the trigger for a fast and spontaneous interaction between the LNPs and the acceptor LUVs, for all tested formulations. This can be rationally explained since the ionizable cationic lipids in the LNP will become protonated at acidic pH and facilitate an electrostatic interaction with the anionic acceptor LUV. The representative cryoTEM images indeed show this clear contact between LNPs and liposomes due to this pH trigger. Due to large overload of lipids coming from the LNP to the LUV, it makes longer evaluation of mixing troublesome. In addition, further timescales might not be very representative since in an actual intracellular scenario there are many more membrane dynamics and protein interactions which cannot be mimicked in our assay. Therefore, instead, we focused on the structural features that appear at the initial interaction between the LNP and LUV. We note that 10PE-LNP-NP1 undergoes a transition from a lamellar to hexagonal at this interface between LNP and LUV, whereas the hexagonal packing of the 49-PE-LNP-NP1 remains in its initial hexagonal configuration. Linking together this observation from cryoTEM with accelerated lipid mixing and improved intracellular silencing demonstrated in our article, together with the widely understood importance of having an intermediate inverse hexagonal structure for oligonucleotide delivery,^{1,2} made us propose the mechanism shown in Figure 6h. The proposal here is that the increased transfection efficiency is mainly due to the bypassing of the initial reconfiguration from lamellar to hexagonal, which makes the process of siRNA release faster and more efficient.

References:

- 1 Semple, S. C. *et al.* Rational design of cationic lipids for siRNA delivery. *Nature Biotechnology* **28**, 172-176 (2010). <https://doi.org:10.1038/nbt.1602>
- 2 Koltover, I., Salditt, T., Rädler, J. O. & Safinya, C. R. An Inverted Hexagonal Phase of Cationic Liposome-DNA Complexes Related to DNA Release and Delivery. *Science* **281**, 78-81 (1998). <https://doi.org:doi:10.1126/science.281.5373.78>